# Cohesin forms fountains at active enhancers in *C. elegans*

Bolaji N. Lüthi[1,2], Jennifer I. Semple [1], Anja Haemmerli[1], Saurabh Thapliyal [3], Kalyan Ghadage[1,2], Klement Stojanovski[4], Dario D'Asaro[5], Moushumi Das[1,2], Nick Gilbert [6], Dominique A. Glauser [3], Benjamin Towbin [4], Daniel Jost [5] & Peter Meister [1] ✉

Transcriptional enhancers must locate target genes with precision. In mammals, topologically associating domains (TADs) guide this process, but the *C. elegans* genome lacks such organization despite containing over 30,000 putative enhancers. Using high-resolution Hi-C, we identify distinct 3D chromatin structures around active enhancers, termed fountains. These ~38 kb cohesin-dependent structures are unique to active enhancers and enriched for topoisomerases and negatively supercoiled DNA, indicating topological stress. Disrupting cohesin collapses fountains and leads to transcriptional upregulation of nearby genes, suggesting fountains act as spatial repressors controlling enhancer–promoter communication. This repression preferentially affects neuronal genes, including *skn-1/Nrf*, which changes isoform usage upon cohesin loss in ASI neurons. Cohesin cleavage also alters nematode movement and foraging behavior, linking 3D genome architecture to neural function and behavior. Thus, fountains represent a distinctive chromatin feature that may ensure enhancer specificity in a TAD-less genome.

Gene transcription is tightly regulated by a combination of promoter-proximal elements and more distant enhancer sequences. Enhancers increase the transcription of target genes by recruiting sequence-specific transcription factors, which in turn attract additional proteins to facilitate transcription. Notably, enhancers are capable of activating transcription independently of their location relative to their target gene, and can act at large distances from the promoter, up to several tens of kilobases in mammals[1].

Recent chromosome conformation capture studies in mammals have shown that enhancer and target promoter are often located in the same megabase-sized three-dimensional domain known as a topologically associated domain (TAD) while TAD segmentation regulates promoter/enhancer contacts and the downstream transcriptional regulation[2–4]. The formation of TADs is the result of chromatin looping,

whereby cohesin extrudes chromatin until it reaches sequence elements bound by the DNA-binding protein CTCF (CCCTC-Binding Factor; for review, ref. 5).

In *Caenorhabditis elegans*, two different studies identified between 19,000 and 30,000 sequences with enhancer-type chromatin features (Fig. 1a, b for L3 stage enhancers[6,7]). These sequences exhibit an open chromatin structure (ATAC-seq peaks), characteristic histone marks (high H3K4 monomethylation coupled to low H3K4 trimethylation), short-stretch bidirectional transcription, and enrichment for initiator sequence element (Inr)[6,7]. A limited number of these sequences were individually tested for their capacity to activate a minimal promoter driving green fluorescent protein (GFP) independently of their location relative to the promoter. This confirmed that these sequences are bona fide enhancers leading to cell- and developmental

[1]Cell Fate and Nuclear Organization, Institute of Cell Biology, University of Bern, Bern, Switzerland. [2]Graduate School for Cellular and Biomedical Sciences, University of Bern, Bern, Switzerland. [3]Department of Biology, University of Fribourg, Fribourg, Switzerland. [4]Organismal Systems Biology, Institute of Cell Biology, University of Bern, Bern, Switzerland. [5]Laboratoire de Biologie et Modélisation de la Cellule, Ecole Normale Supérieure de Lyon, CNRS, UMR5239, Inserm U1293, Université Claude Bernard Lyon 1, Lyon, France. [6]Medical Research Council Human Genetics Unit, Institute of Genetics and Molecular Medicine, University of Edinburgh, Edinburgh, UK. ✉e-mail: peter.meister@unibe.ch

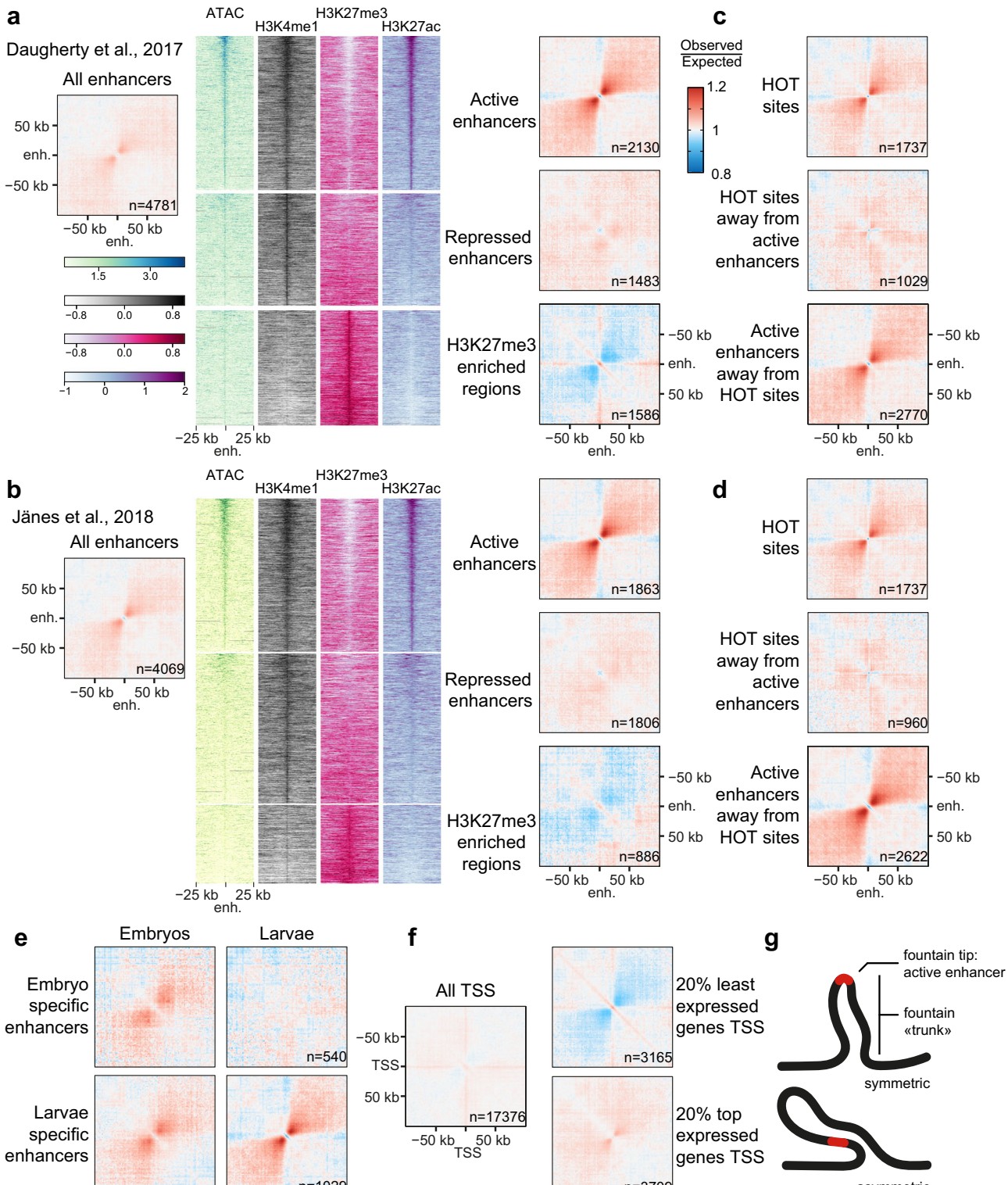

**Fig. 1 | Active enhancers colocalize with loose 3D structures or fountains.**
**a** Average contact frequency maps centered on annotated enhancers from Daugherty et al.[7], highlighting the formation of 3D structures. Further segmentation into active or repressed enhancers and H3K27me3-covered regions highlight the specificity of fountains for active enhancers. **b** Average contact frequency maps centered on annotated putative enhancers (all stages) from Jänes et al.[6], further segmented using histone marks as in Daugherty et al. Highly Occupied Target (HOT) sites are not forming fountains. Average contact frequency maps for all HOT sites, for HOT sites located more than 6 kb away from active enhancers and for active enhancers located more than 6 kb away from HOT sites, using active

enhancers from Daugherty et al. (**c**) or Jänes et al. (**d**). **e** Fountains for active enhancers are specific for their developmental stage. Average contact maps for embryo-specific and L3 larvae-specific ATAC-seq peaks (ATAC-seq peaks in the top 3 deciles at the considered stage and in the bottom 3 deciles at the other stage). **f** Transcription start sites (TSS) do not colocalize with fountains. Average contact frequency maps centered on all TSS, and TSS segmented by gene expression. Small fountains present in the most expressed 20% genes are most likely due to active enhancers located in the vicinity of the TSS. **g** Cartoon of a typical fountain, with the active enhancer located at the tip of the fountain structure, both a symmetric and asymmetric fountain are shown.

stage-specific expression of the transgene. It is currently unclear whether and how the activity of these enhancers is limited to their target genes, as unlike mammals, nematodes do not harbor megabase-sized TADs on their autosomes, although smaller, kilobase-sized compartments similar to A and B compartments can be identified in Hi-C data[8]. Additionally, no CTCF homolog or functional homolog has been identified in the *C. elegans* genome, and no nematode boundary elements have been described to date. Moreover, the primary long-range loop extruder in nematodes is not cohesin as in mammals, but condensin I[9]. Taken together, these observations suggest that cohesin function is largely divergent in nematodes compared to mammals and that an alternative mechanism to TADs might regulate enhancer-promoter contacts and the ensuing gene expression. Hereafter, we show that active nematode enhancers are located at the tip of small-scale 3D structures, which we call fountains and similar to described jets[10], plumes or flares[11–13], created by cohesin activity. Cohesin[COH-1] cleavage leads to upregulation of active enhancer- and fountain-proximal genes, in particular genes expressed in neurons. Strikingly, cohesin[COH-1] cleavage leads to a breadth of behavioral changes, linking 3D genome organization by cohesin, neuronal gene expression and nervous system function in animal behavior.

## Results

### Active enhancer loci correlate with 3D fountains

To assess whether enhancers would colocalize with specific three-dimensional genomic features, we generated average chromatin conformation capture contact maps for third larval stage animals centered on enhancer sequences[6,7]. These contact maps revealed a clear and distinct enlarged second diagonal perpendicular to the main Hi-C diagonal, extending several kilobases from the enhancers (Fig. 1a, b). These increased contact probabilities between the enhancers and neighboring sequences are indicative of the formation of a loose loop–like structure centered on the enhancer sequences, resulting in the partial alignment of two branches of the loop either side of the enhancers (Fig. 1g). Hereafter, we call these structures fountains, in line with other manuscripts describing these structures[12,14,15].

Enhancers were further characterized in one of the studies above using ChromHMM genome segmentation[7]. This division classifies enhancers into active enhancers covered with H3K4me1 and H3K27ac, repressed enhancers with H3K4me1, weak H3K27me3, and no H3K27ac, and H3K27me3-enriched regions covered with H3K27me3, deposited by the PRC2 Polycomb complex (Fig. 1a). When we averaged the contact maps of the different enhancer types and the H3K27me3-enriched regions, we observed that active enhancers were associated with fountains, whereas repressed enhancers did not show any fountains, and H3K27me3-covered regions showed a decrease of expected contacts in the second diagonal, as well as a cross-like high contact probability feature, suggesting that H3K27me3 regions cluster together in vivo (Fig. 1a). Similarly, when we segmented the second enhancer dataset[6] based on the same ChromHMM chromatin segmentation[7], we made almost identical observations: active enhancers were associated with fountains, while either repressed enhancers or trimethylated H3K27-covered regions were not (Fig. 1b).

In previous ChIP studies, a set of genomic regions bound by multiple transcription factors was identified, named Highly Occupancy Target (HOT) sites[16,17]. As enhancers are binding sites for transcription factors, HOT sites and enhancers often co-occur on nearby loci. To investigate whether HOT sites, active enhancers, or both types of sequence elements colocalize with fountains, we created average contact maps centered on HOT sites that either overlapped with active enhancers or were distinct from them. Our results indicate that HOT sites only correlated with fountains (Fig. 1c, d) when these overlapped with active enhancers. In contrast, HOT sites not overlapping with active enhancers did not create fountains, while active enhancers not overlapping with HOT sites did (Fig. 1c, d).

We next wondered whether fountains would be stage-specific, by selecting enhancers active in embryos and inactive in the third larval stage, or conversely. We calculated average chromatin conformation contact maps for these different enhancer sets at different stages using our Hi-C data of third larval stage and publicly available Hi-C data performed with the same protocol in embryos[18]. For both enhancer sets, we observed clear fountains in their respective developmental stages (Fig. 1e). In contrast, no fountains could be observed for enhancers active only in embryos using larvae Hi-C maps, and larvae-specific enhancers created smaller and weaker fountains in embryonic Hi-C data (Fig. 1e). The latter might be due to the fact that mixed stage embryos were used to perform Hi-C, in which a variable proportion of animals are already in late developmental stages during which some larval enhancers are already active. We conclude that fountain formation is stage-specific, in agreement with the activity of the enhancers.

Finally, we explored whether fountains were unique to enhancers or if they were associated with other open chromatin regions such as active transcription start sites (TSS). The nematode genome is highly compact and enhancers are located only a couple kilobases away from their putative target promoters[6,7]. To examine TSS-specific 3D structures, we calculated the average contact maps of the bottom or top 20% of all expressed genes ranked by expression level. We found no evidence of fountain formation in the bottom 20%, while the 20% most expressed genes showed very limited fountain formation (Fig. 1f), vastly smaller than the fountains observed at active enhancers (Fig. 1a, b). In summary, our data shows that fountain formation is a feature specific to active enhancers and not selectively associated with transcribed genes or HOT sites. We envision fountains as loose loop-like structures with the active enhancer sitting at the tip of the loop (Fig. 1g).

### Enhancers are binding sites for cohesin[COH-1]

Recent studies have described similar structures orthogonal to the Hi-C diagonal in bacteria[19], *C. elegans*[14], fungi[13], T cells[10] and zebrafish sperm[11], as well as during zygotic genome activation (ZGA) in zebrafish[15]. These structures are thought to be formed through bidirectional loop extrusion by Structural Maintenance of Chromosomes (SMC) complexes repeatedly loaded at defined loci (in our case, the active enhancers). Nematodes express five different SMC complexes in the soma: canonical condensin I and II, an X-chromosome-specific condensin I variant, and two variants of cohesin, differing by their kleisin subunit, SCC-1 or COH-1. We previously demonstrated that condensin I performs long-range loop extrusion during interphase (>100 kb), while condensin II has no interphasic function[9]. Cohesin[SCC-1] is involved in sister chromatid cohesion during mitosis and exclusively expressed in dividing cells[20]. Early immunofluorescence studies showed that COH-1 is expressed in all cells, suggesting cohesin[COH-1] is the major interphasic cohesin. In our previous study, we showed that the common SMC-1 subunit of cohesin is expressed ubiquitously at high levels[9] (Fig. 2a), suggesting cohesin[COH-1] is present in most, if not all cells. Direct quantification of COH-1 and SCC-1 abundance using identical tags on the two kleisin subunits showed that in entire animals, COH-1 is 6 times more abundant than SCC-1[9]. To determine whether enhancer sequences were enriched for cohesin[COH-1], we used mod-ENCODE ChIP-seq data[17]. We observed that cohesin[COH-1] is specifically enriched in a broad region centered on active enhancers, extending several kilobases away from the enhancer itself (Fig. 2b, upper part, red line and heatmap). In contrast, repressed enhancers had only a small enrichment limited to the enhancer sequences (green line). Cohesin[COH-1] was slightly depleted on H3K27me3-covered regions compared to neighboring sequences (blue line). To further investigate whether cohesin[COH-1] enrichment on active enhancers correlated with the size of the fountains, we classified active enhancers into 5 classes based on their cohesin[COH-1] enrichment and created average contact

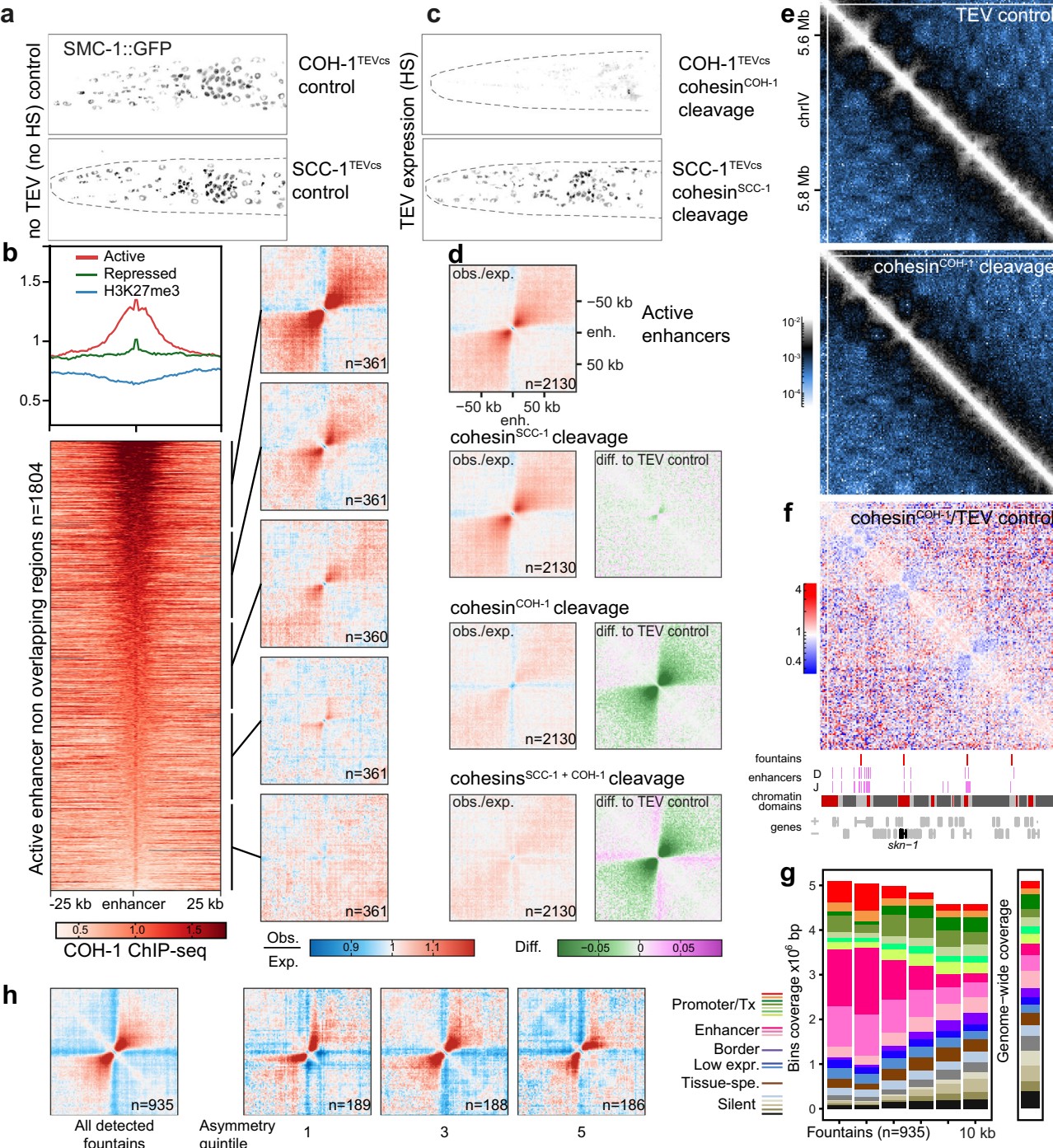

**Fig. 2 | Cohesin^COH-1 creates fountains at active enhancers. a** Fluorescence signal in the head of third larval stage control animals expressing SMC-1::GFP without TEV expression. The light patch in each nucleus corresponds to the nucleolus. Strains in which TEV cut sites have been inserted in either of the two alternative cohesin kleisins COH-1 and SCC-1 are shown. **b** (top left) Average COH-1 ChIP-seq profiles in young adult animals at active (red) or repressed (green) enhancers and H3K27me3-enriched regions (blue). (bottom left) Heatmap of COH-1 enrichment on active enhancer regions, sorted by COH-1 ChIP-seq enrichment. (left) Average contact frequency maps centered on enhancers segmented according to COH-1 ChIP-seq abundance. **c** SMC-1::GFP signal as in a upon cleavage of COH-1 and SCC-1. **d** Cleavage of cohesin^COH-1 but not cohesin^SCC-1 leads to disappearance of active enhancer fountains. Average contact frequency maps, centered on active enhancers upon cleavage of COH-1 and/or SCC-1. The difference between the kleisin

cleavages and TEV control is shown on the right with a green/magenta color scale. **e** Contact frequency map in TEV control animals (top) and upon cohesin^COH-1 cleavage (bottom) in a section of chromosome IV. **f** (top) Ratio contact frequency map between cohesin^COH-1 cleavage and TEV control, highlighting the disappearance of the fountains (blue, negative values). (bottom) Tracks showing detected fountains, active enhancers (Daugherty et al.[7]: D; or Jaenes et al.[6]: J), chromatin domains[23] and location of genes (bottom). **g** Autosomal chromatin state coverage of detected fountain bins (left) and their adjacent bins, sorted by distance to the detected fountains. Autosome-wide coverage is shown on the right side. Chromatin states from ref.[23]. **h** Average contact frequency maps centered on detected fountains for all fountains and for top, middle and bottom asymmetry quintiles of all fountains, based on the asymmetry score described in the methods section.

frequency maps for each class. Our analysis revealed that active enhancers with high cohesin[COH-1] enrichment generated large fountains extending several tens of kilobases away from the active enhancer locus (Fig. 2b, right side), while active enhancers of the second-to-lowest quintile formed very small fountains, and active enhancers of the lowest cohesin[COH-1] ChIP-seq enrichment quintile did not correlate with fountains.

To directly investigate the role of cohesin[COH-1], we made use of our previously described inducible cleavage system for the cohesin kleisins COH-1 and/or SCC-1[21]. When Tobacco Etch Virus (TEV) protease expression was induced during the first larval stage (3 h after diapause exit), cleavage of COH-1 (89% cleaved[9]), but not of SCC-1 caused the disappearance of most GFP-tagged SMC-1[9] by the third larval stage (Fig. 2c, quantified in ref. 9), indicating that COH-1 cleavage leads to the degradation of the entire cohesin[COH-1] complex and confirming that cohesin[COH-1] is the major cohesin isoform. We calculated average contact maps centered on active enhancers in control animals and upon cleavage of either or both cohesin kleisins. Cleavage of SCC-1 only marginally modified contact maps, in contrast to the cleavage of COH-1, in which fountains were almost completely absent (Fig. 2d, e). Similar results were obtained upon simultaneous cleavage of COH-1 and SCC-1 (Fig. 2d). Cohesin[COH-1] is thereby necessary for the maintenance of fountains at active enhancers. Given the loop extrusion activity of cohesin[22], the most likely model for the formation of the fountains is the specific loading of the cohesin complex at active enhancers followed by bidirectional loop extrusion randomly stopping on either side, as suggested by polymer modeling[15].

## Cohesin-dependent fountains colocalize with active enhancers

The effects of cohesin[COH-1] cleavage can be visualized by creating contact-ratio maps between cleaved and control contact maps (Fig. 2e, ratio in f). We used these ratio maps to locate fountains, as changes in contact frequency are most likely direct effects of cohesin[COH-1] cleavage. We filtered the diagonal of the ratio maps with a model-based kernel similar to the average fountain shape observed around enhancers, leading to a fountain similarity score for each locus along the genome (Fig. S1, Methods section and Supplementary Data 1). Using this approach, we identified 935 fountains in the entire genome, whose tips are the local optima of the similarity score with significant prominence ("fountain score"; Supplementary Data 2, 3 for the location and a gallery of all detected fountains). For each fountain, we measured its length, corresponding to the typical extension of the fountain around the tip (i.e., the typical loop size extruded by cohesins loaded at the tip), and its (a)symmetry, corresponding to possible deformations in fountain shapes along 5′ or 3′ sides of the tip (i.e., asymmetries in cohesin progression around the tip; see Methods and Fig. S1a–f for details and validation of the approach, as well as a comparison with fontanka[15]). Using this method, we determined that the size of 80% of the fountains lies between 22 and 78 kb, with a median fountain length of 38 kb, but some larger fountains could still be detected at 150 kb (Fig. S1j). Notably, while most fountains were symmetric (Fig. S1g), a significant number of fountains were asymmetric, with more contacts on one side than on the other, implying that loop extrusion by cohesin[COH-1] might be directionally biased in those cases (Fig. 1g). Asymmetry correlated weakly with the abundance of enhancer type chromatin states, enriched on the side where the fountains were longest (Fig. S1i).

Regarding their chromosomal location, fountains did not show any preference, neither for the center nor the arms of chromosomes (Fig. S2a–d), yet they were slightly depleted on chromosome IV and V compared to chromosomes I to III. We did not observe any particular preference for fountain location relative to the boundaries of the X chromosome TADs in dosage-compensated hermaphrodite animals (Fig. S2e). We next compared the location of fountains with ChIP-seq data. It is important to notice here that the resolution of Hi-C data

(2 kb) is orders of magnitudes lower than ChIP-seq data. We found that COH-1 was enriched at fountains and positively correlated with fountain score ($R = 0.52$, Fig. S1k). Chromatin states at fountain tips, as well as those of the neighboring bins were enriched for enhancer states compared to genome-wide coverage (Fig. 2g), further supporting the notion that fountains are related to enhancers. When we compared the location of detected fountains relative to the different types of enhancers, we found that fountain tips are significantly closer to active enhancers than to repressed ones or H3K27me3-covered regions (Fig. S3a, b). Similarly, fountain tips and their ±2 kb flanking regions more often contained multiple active enhancers when compared to equally sized control regions located at equal distance between fountain tips, with up to 6 or 11 active enhancers located at the fountain tips and the two neighboring bins, depending on the enhancer mapping study[6,7]. In contrast, repressed enhancers or H3K27me3-covered regions were not enriched in fountain tips and their flanking regions (Fig. S3c, d). To further test if fountains preferentially form at clustered active enhancers, we divided the whole genome into 6 kb bins and counted the number of active enhancers overlapping with each bin. While fountains represent only 5.6% of all genomic 6 kb bins, fountain tip bins were more likely to overlap with 6 kb genomic bins encompassing larger numbers of active enhancers. Accordingly, fountain tip bins are vastly overrepresented in genomic bins clustering three or more active enhancers (Fig. S3e, f). Since we did see many fountains at bins with more than one enhancer, it seems likely that fountain strength, and therefore detection, improves with the number of enhancers clustered together. Indeed, we saw a clear correlation between the number of active enhancers per fountain bin and the fountain prominence score produced by our detection algorithm (Fig. S3g, h). Together, we demonstrated that fountains identified on differential Hi-C maps and dependent on cohesin[COH-1] integrity are primarily associated with active enhancers, while their prominence correlates with the number of clustered enhancers.

A previous study segmented the nematode genome based on histone marks and ATAC-seq data into active, regulated and border domains[23]. We therefore asked whether fountain tips would preferentially colocalize with one type of chromatin domain. Indeed, active domains were enriched at fountain tips, as well as on either side of them (Fig. S4a). Conversely, fountains located in active domains were significantly more prominent than fountains located in border or regulated domains (Fig. S4b). These features, although identified genome-wide, were also apparent when considering individual loci (Figs. 2f, S4c, and S11a).

In summary, the genome-wide identification of the locations of cohesin[COH-1]-dependent fountains on Hi-C maps revealed a robust correlation between fountains and active enhancers, as well as COH-1 enrichment. Furthermore, there is a slight correlation between fountain asymmetry and unequal transcriptional activity and/or enhancer chromatin state between the two branches of the loop that emanate from the fountain tips.

## Active enhancers and fountains are bound by topoisomerases

To further understand fountain formation, we examined published ChIP-seq data at enhancer sequences acquired at the same developmental stage[24]. Bidirectional transcription at enhancers results in the production of short transcripts, and as expected, RNA polymerase II is enriched at enhancer sequences (Fig. 3b). In contrast to cohesin[COH-1], which extends several kilobases away from the enhancer itself along the entire length of the fountains, RNA polymerase II is only enriched at the tip of the active enhancers (Fig. 3a, b; refs. 6,24). Interestingly, we found that the enrichment of RNA polymerase II at active enhancers correlated with the abundance of COH-1 (Spearman correlation coefficient $R = 0.63$), which in turn determined fountain sizes (Fig. 2b). These findings prompted us to explore the interplay between enhancer transcription and fountain formation.

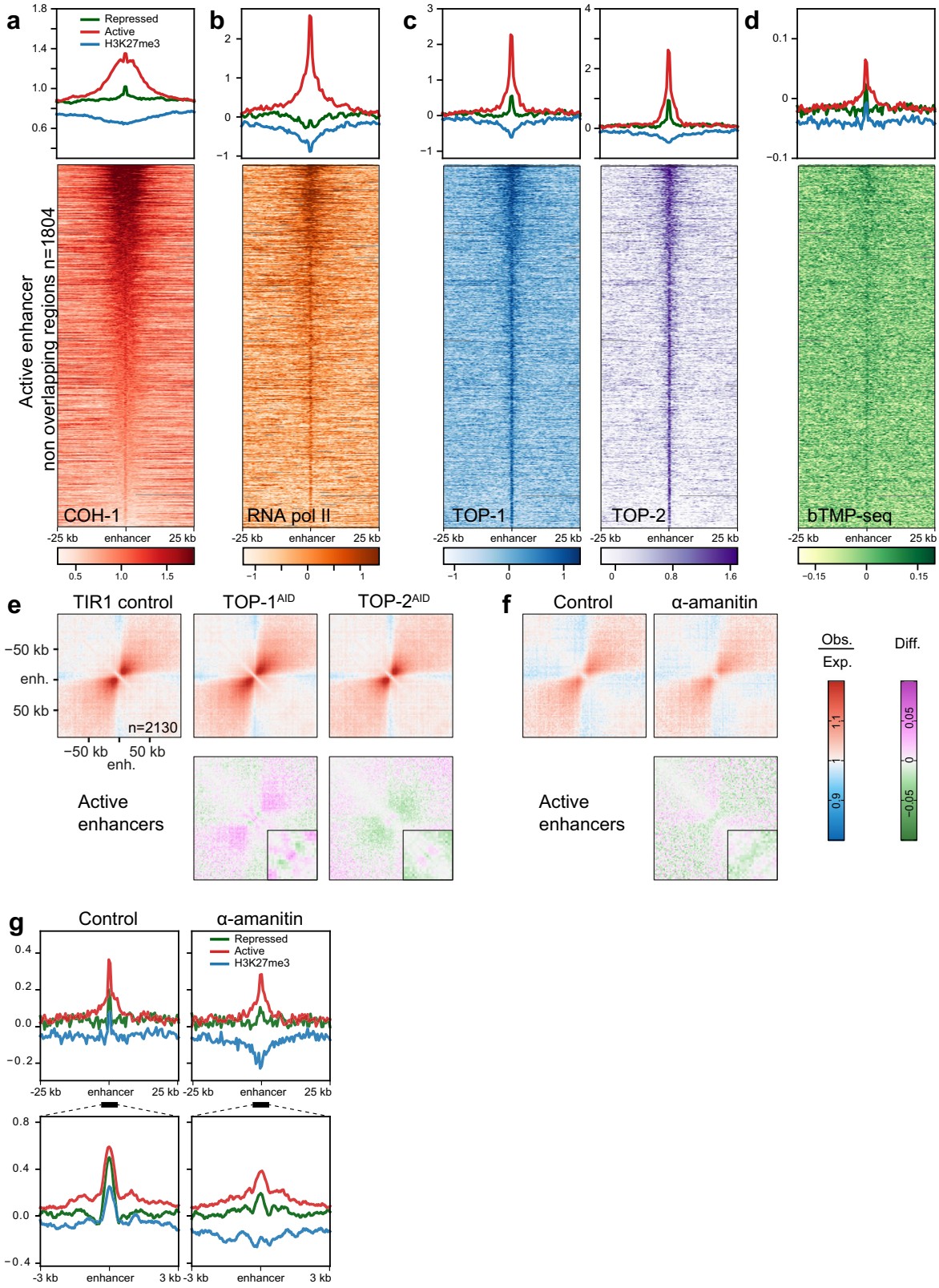

During transcription, the opening of the double helix and movement of the RNA polymerase generates negative supercoils behind the polymerase, and in the case of divergent transcription at enhancers, negative supercoils accumulate between the two polymerases progressing in opposite directions. In front of the polymerases, positive supercoils are generated, in other words, on either side of the bidirectionally transcribed enhancer sequences. Topoisomerases, the enzymes which relax supercoils by cleaving and unwinding DNA, are therefore expected to be present at active enhancers. Conversely, loop extrusion by cohesin has been shown to modify the topological state of DNA in vitro[25]. Indeed, a clear, specific and correlated enrichment for both TOP-1 and TOP-2 was observed at active enhancers (Fig. 3c, $R = 0.83$). Additionally, for both topoisomerases, the enrichment correlated with cohesin[COH-1] abundance ($R = 0.6$ and $0.59$, respectively). As

**Fig. 3 | Active enhancers are bound by topoisomerase and enriched for the negative supercoil binder bTMP.** (top) Average ChIP-seq profiles at active or repressed enhancers as well as H3K27me3-enriched regions for **A.** COH-1 (young adults), same as in 2b for reference. **B** RNA pol II (L3), **C** (left) TOP-1 (L3) and (right) TOP-2 (L3). (bottom) **D** (top) Average bTMP-seq profiles in L3 at active or repressed enhancers and H3K27me3-enriched regions, (bottom) Heatmap of bTMP binding in active enhancer regions, sorted by COH-1 ChIP-seq enrichment. **E** Depletion of topoisomerases slightly alters fountain strength, especially at the tip. Average contact frequency maps and difference to control, centered on active enhancers. (left) TIR1 control, (right) TOP-1 or TOP-2 auxin-mediated degradation. (bottom)

The difference between the depletion of the topoisomerases and the control, (inset) zoomed in at the fountain tip. **F** Inhibition of transcription with α-amanitin slightly alters fountain strength, especially at the tip. Average contact frequency maps and difference to control centered on active enhancers. (left) control, (right) α-amanitin inhibited transcription. (bottom) The difference between transcription inhibition and the control, (inset) zoomed in at the fountain tip. **G** Average bTMP-seq profiles upon transcription inhibition in L3 (top) centered at active, or repressed enhancers, as well as H3K27me3-covered regions with 25 kb flanking regions (500 bp bin size), (bottom) 6 kb window centered on enhancers (20 bp bin size).

for RNA polymerase II, the TOP-1 and TOP-2 ChIP-seq signal was limited to the enhancer sequences at the tip of the fountains. These findings suggest that topoisomerases are regulating the supercoiling at enhancer sequences created by repeated small-stretch transcription. To directly test whether enhancer sequences are indeed negatively supercoiled, we performed biotinylated psoralen crosslinking followed by sequencing (bTMP-seq[26]). bTMP intercalates preferentially with negatively supercoiled DNA and can be crosslinked to DNA using long-wavelength UV irradiation. As for topoisomerases, bTMP was enriched in the tip of the fountains at active enhancer sequences, demonstrating that these sequences are negatively supercoiled in vivo (Fig. 3d). Importantly, similar enrichments were observed at the detected fountain tips when plotting the abundance of COH-1, RNA pol II, TOP-1/-2 or bTMP across the 935 fountains (Fig. S5a–d).

To further investigate the role of topoisomerases in the maintenance of fountains, we created average contact frequency maps centered on active enhancers from topoisomerase depletion data (Fig. 3e; ref. 24). In these experiments, GFP- and degron-tagged topoisomerases were depleted by a 1 h treatment with auxin, leading to the disappearance of the nuclear GFP signal and uniform depletion of the ChIP-seq signal across the genome[24]. As for our data, average contact maps in control animals showed clear fountains (Fig. 3e, TIR1 control). Upon depletion of either TOP-1 or TOP-2, fountains were only mildly affected and changes impacted mostly the tip of the fountains which show a high enrichment for both enzymes (Fig. 3e, TOP-1[AID], TOP-2[AID], inset). TOP-1 depletion led to a slight increase in contacts along the trunk of the fountain, while the contacts on either side of the active enhancers were slightly reduced. Conversely, TOP-2 depletion resulted in a slight decrease in contacts along the fountain trunk, with a limited increase in contacts on either side of the active enhancers (Fig. 3e, TOP-1[AID], TOP-2[AID], difference maps to TIR1 control). We conclude that short-term depletion of either TOP-1 or TOP-2 has only a marginal effect on fountains, mainly at the tip.

To further characterize the role of transcription in fountain maintenance, we blocked transcription for 5 h using α-amanitin. We first investigated whether transcription inhibition was accompanied by altered supercoiling by performing bTMP-seq and plotted the average enrichment at different enhancer types upon α-amanitin treatment (Fig. 3g, 25 kb (top) and 6 kb (bottom) windows centered on enhancers). As expected, since transcription inhibition would decrease supercoiling, α-amanitin treatment lowered bTMP enrichment on all enhancer types, more strikingly at the H3K27me3-covered regions, although the reason for this remains unclear. At active enhancers, bTMP enrichment was decreased at the enhancers but slightly higher on either side of them, suggesting that the absence of transcription would lead to the relocation of negative supercoils away from the enhancer sequences, but not completely abolish these. Additionally, we performed Hi-C upon transcription inhibition and created average contact maps centered on active enhancers. Similarly to topoisomerase depletion, α-amanitin treatment only slightly decreased fountain strength (Fig. 3f, difference map to control). Collectively, our findings led us to the conclusion that fountains are enriched for RNA polymerase II, topoisomerases and bTMP, indicating probable DNA

supercoiling. However, the presence of either RNA polymerases or topoisomerases is not essential for the maintenance of the fountains, indicating that these structures remain stable for at least 1 h without topoisomerases and 4 h without transcription once they are formed, most likely stabilized by the presence of cohesins.

## Cohesin[COH-1] cleavage correlates with transcriptional activation of genes close to active enhancers and fountain tips

We previously analyzed the transcriptional consequences of COH-1 cleavage in entire animals[9]. Analyzing this data in isolation with less stringent filtering criteria, 895 genes were significantly upregulated, and 993 genes were significantly downregulated ($p < 0.05$). Most changes in transcript abundance were small, with only 98 up- and 52 down-regulated genes with a fold change larger than 1.41 ($|\log2FC| > 0.5$). As ATAC-seq, modified histone ChIP-seq and RNA-seq are made on entire animals with a large number of different cell types, the target gene of each enhancer has been determined only for a handful of enhancers[6,7]. We therefore distributed genes into six categories by assigning each enhancer to the closest TSS (or to none if the gene was not the closest TSS to any enhancer, Fig. 4a). "Active", "Repressed", "H3K27me3-covered" categories were TSS of genes closest to only one type of enhancer; "Mixed active and repressed" category were TSS of genes closest to a mixture of enhancers from the active and either of the two categories - repressed or H3K27me3-covered; and "mixed repressed" were genes closest to a mixture of repressed enhancers and H3K27me3-covered regions. We then analyzed the impact of COH-1 cleavage on these different gene categories (Fig. 4a). When no enhancer was present in the vicinity of the gene (the largest number of genes), average expression after cleavage of COH-1 was slightly lower than in the control experiment. For genes proximal to mixed or active enhancers, the change in expression levels of these genes was biased towards higher expression (Fig. 4a, red and violet boxes). In contrast, genes proximal to H3K27me3-covered regions were evenly distributed between up- and down-regulated (Fig. 4a, green box). Therefore, genes close to active enhancers were upregulated upon COH-1 cleavage, suggesting fountain formation limits active enhancer activity. The fact that genes close to a mixture of active and repressed enhancers and even a mixture of the two repressed categories were also biased towards upregulation, suggests a role for cohesin[COH-1] in regulating complex transcriptional landscapes, though the number of genes in these groups is small.

We next analyzed transcriptional changes upon COH-1 cleavage based on the location of the detected fountains (Fig. 2c). For this, we first measured the distance between genes and the closest fountain and split genes based on their transcriptional modulation upon COH-1 cleavage. We found that the average distance to fountain tips of non-changing or downregulated genes was not significantly different, while upregulated genes were significantly closer to fountain tips (Fig. 4b). Second, we focused on fountain tips and compared those to a set of control regions located at the midpoint between neighboring fountain tips (Fig. 4c). Genes that overlap fountain tips were significantly biased towards upregulation whereas genes overlapping control bins were equally distributed between up- and down-regulation (Fig. 4d). As

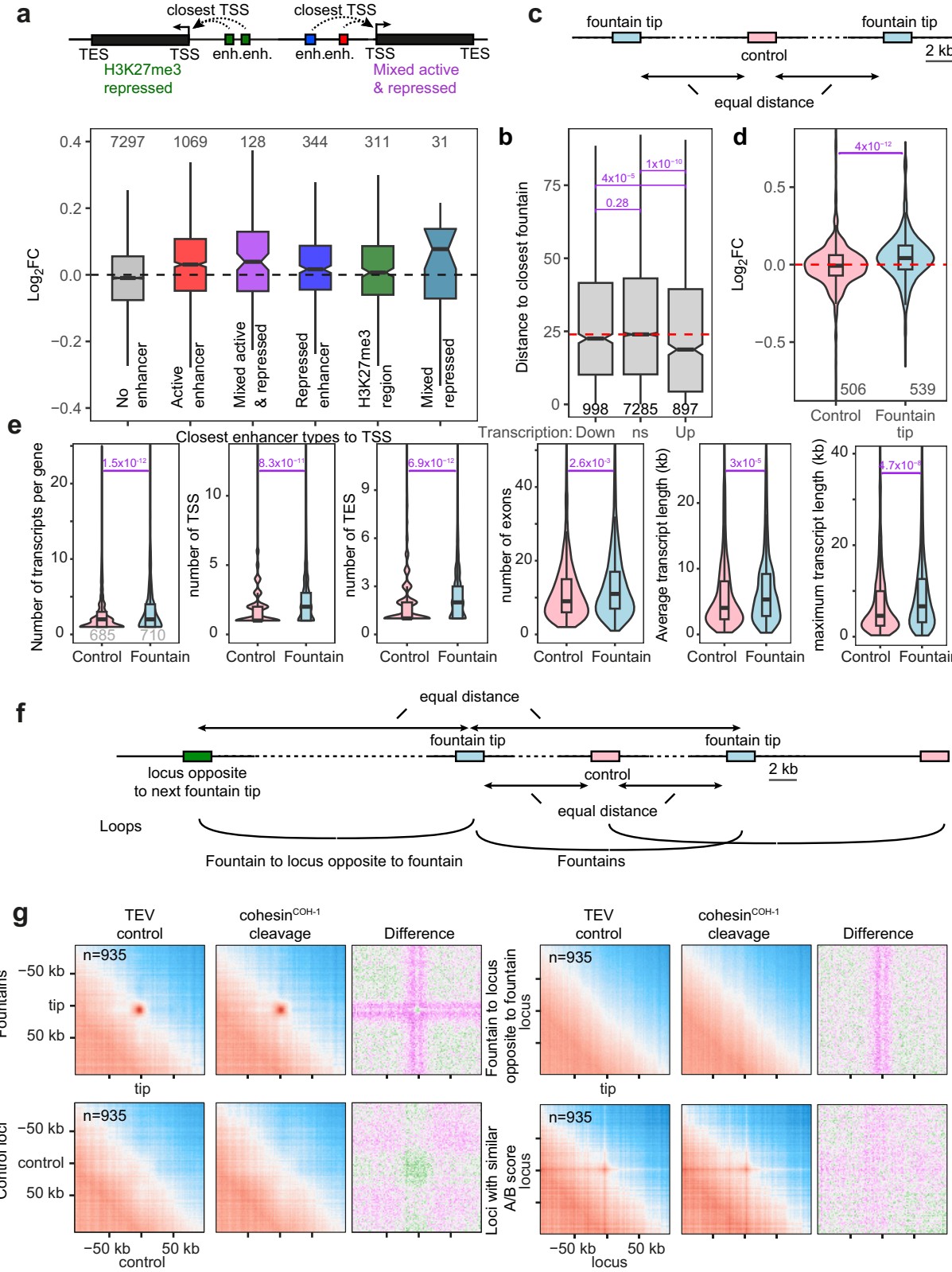

fountains are enriched for cohesin[COH-1], gene upregulation upon COH-1 cleavage might be a consequence of cohesin[COH-1] enrichment, independently of the fountains themselves. To disentangle cohesin[COH-1] enrichment from fountain proximity, we compared gene expression changes in fountain tip bins (±2 kb) *versus* genomic bins with similar cohesin[COH-1] enrichment to fountain tips that were located more than 30 kb away from these (Fig. S6a). Upon cohesin[COH-1] cleavage, fountain

tip genes were significantly upregulated whereas a sample of non-fountain bins, similarly enriched for cohesin[COH-1] were not (Fig. S6b–d). Repeating a hundred times the same experiment by picking different fountain tips/non-fountain tip bins led to the same result (Fig. S6e). Together, this demonstrates that fountain ablation by COH-1 cleavage upregulates fountain tip-proximal genes in a way that is specific to these 3D structures. Therefore, cohesin's repressive effect on fountain-

**Fig. 4 | Active-enhancer proximal genes are upregulated upon COH-1 cleavage, fountains cluster in the nuclear space. a** Genes were grouped according to the type of L3 stage enhancers from Daugherty et al. (2017) that their TSS was closest to. The log2 fold change of all expressed genes upon cleavage of COH-1 are plotted according to enhancer type. **b** Boxplot of the distance of genes from the nearest fountain tip bin. Genes were grouped by their change in expression upon COH-1 cleavage: up and down regulated or not changing significantly (NS). The dashed red line indicates the median distance to the nearest fountain tip bin of NS genes. In **b**, **d**, **e**, the number of genes in each group is indicated at the bottom of the plot, and adjusted *p*-values from a two-sided Wilcoxon rank sum tests are shown in purple. **c** Schematic representation of regions used in (**d**, **e**). 2 kb fountain tip bins were compared with control bins of the same size located at the midpoint between neighboring fountains. **d** Log2 fold change of all expressed genes upon COH-1$^{cs}$

cleavage that overlap control bins (506 genes) and fountain-tip bins (539 genes). **e** Comparison of features of genes that overlap with control and fountain-tip bins. **f** Graphical depiction of the loops analyzed in (**g**). **g** Top row, left: average contact maps show loops between pairs of fountain tips under TEV control conditions. No such loops were observed for control loci (second row, left), or between fountain and loci located at the similar distance but on the opposite side to the next fountain (first row, right), or between loci with similar A/B score than fountains (second row, right). Average contact maps are shown both for the TEV control and upon cohesin$^{COH-1}$ cleavage, conditions, as well as the differential map (cleavage/control) - same color scales as in Fig. 2d. The midline, boundaries and whiskers of the box-plots in panels a-d show the median, 25th and 75th percentiles, and the minimum/maximum value no further than ±1.5× the interquartile range (IQR), respectively. Notches (in **a**, **b**) indicate ±1.58 IQR/sqrt(*n*).

tip proximal genes is not simply a consequence of high levels of cohesin randomly colliding with the transcriptional machinery.

Interestingly, fountain tips and their neighboring bins contain genes more highly expressed than the average of all genes in the 40 kb surrounding the fountain tip (Fig. S7a–c). Asymmetric fountains show a slightly higher gene expression in the tip-adjacent bin on the side where the fountains are shortest (Fig. S8a), and fountain disappearance upon COH-1 cleavage did not affect this asymmetry in relative transcription levels (Fig. S8b), suggesting that highly expressed genes could block cohesin$^{COH-1}$ loop extrusion and be the cause, not the consequence of fountain asymmetry. Changes in gene expression upon cohesin$^{COH-1}$ cleavage had a limited correlation with the number of enhancers at the fountain tip: considering 6 kb regions overlapping fountain tips and their two neighboring bins, there is an increase in log2 fold change upon COH-1 cleavage as the number of enhancers in those bins increases from 0 to 1 to 2, however after that, the linear relationship breaks down (Fig. S7d, e). Additionally, genes overlapping fountain bins were more complex than genes overlapping control bins: they had a significantly greater number of transcripts, TSSs and Transcription End Sites (TES) per gene, a larger number of exons, and both their average and maximum transcript length was greater (Fig. 4e), indicating that fountains could be necessary for the regulation of complex transcriptional landscapes.

### Fountain tips cluster in 3D hubs inside the nuclear space

Active enhancers exhibit a well-documented tendency to spatially cluster, forming regulatory hubs enriched in transcription factors, Mediator, and coactivators that cooperatively regulate gene expression[27,28]. To assess whether fountain tips harboring active enhancers similarly engage in higher-order clustering in *C. elegans*, we quantified average contact probabilities between cis-located fountain tips and corresponding control regions as outlined above (Fig. 4f). Our analysis revealed a marked and specific enrichment of contacts between fountain tip pairs, in contrast to control regions or to equidistant loci situated on the opposite flank of adjacent fountains (Fig. 4g). Notably, loci with matched A/B compartment scores and equidistant spacing relative to fountain tips exhibited only modest levels of interaction, consistent with weak chromatin state–driven clustering (Fig. 4g). These findings suggest that fountain tips cluster within the nuclear space, forming subnuclear fountain domains. To interrogate the role of cohesin in modulating fountain contacts, we examined changes in average contact frequencies following cleavage (Fig. 4g, cohesin$^{COH-1}$ cleavage). Contact frequencies between fountain tips slightly decreased, whereas contacts between fountain trunks and neighboring loci increased, leading to a cross-like pattern on the differential contact frequencies map. This redistribution of contact patterns extended into regions flanking the fountain tip by approximately ±14 kb, indicating a localized architectural perturbation. The observed pattern strongly supports a model in which cohesin-mediated loop extrusion forms spatially insulated fountain domains, with tips acting

as hubs for enhancer clustering. Upon cohesin cleavage, these looped architectures disassemble, leading to slight contact depletion at tips and ectopic interaction enrichment along the linear genome, consistent with loss of insulation and increased trunk promiscuity.

To better understand the effect of cohesin$^{COH-1}$ cleavage, we analyzed enhancer-promoter contacts at the HiC fragment level in order to increase resolution. We ranked promoters based on their linear distance to the nearest active enhancer and then compared the contact frequency ratios between cohesin$^{COH-1}$ cleavage and control conditions, alongside changes in transcriptional activity of the corresponding gene. Independent of the chosen set of active enhancers[6,7], the contact ratios were centered around 1 for promoters closest to the active enhancer (rank 1), but slightly decreased for promoters at greater distances (ranks 2–5) (Fig. S9a, b). Gene expression changes following cohesin$^{COH-1}$ cleavage mirrored the results obtained using binned Hi-C data. Genes with promoters closest to an active enhancer were upregulated, and upregulation decreased as the distance to the enhancer increased (Fig. S9c, d). Overall, we could find no simple association between changes in contact frequency and the direction of gene expression changes. This is likely due to the inherent limitations of whole-animal Hi-C and mRNA-seq data, which lacks the resolution to distinguish between different cell types and small changes in genomic distances. Future studies using higher resolution techniques could provide more insights into this relationship.

### Genes upregulated upon COH-1 cleavage are mostly neuronal

Enhancers are believed to drive cell-type specific expression, and indeed, putative nematode enhancers whose activity was individually tested in animals show cell-type specific expression[6,7]. We reasoned that if COH-1 cleavage led to mild upregulation of genes proximal to active enhancers, this might be due to cell-type-specific expression of the latter genes, as RNA-seq is performed on entire animals. Small overall changes would be observed even if transcription levels in individual cells experience major modulation if these genes are expressed only in a few cells per animal. To interrogate whether genes with changed expression levels would belong to a specific cell type, we used two different approaches to analyze the list of significantly up- and down-regulated genes[29]. First, we performed Tissue Enrichment Analysis (TEA) and GO term enrichment, which allows us to determine in which cell type a gene set is most likely expressed. Genes upregulated upon COH-1 cleavage showed a striking enrichment for neuronal cell types, as well as GO terms and phenotypes associated with neurons (Fig. S10a, b). In contrast, downregulated genes were associated with a range of non-neuronal tissues, in particular germline cells and core cellular and metabolic processes (Fig. S10a, b). Second, we asked in which cells genes significantly up- or down-regulated were expressed using previously published single-cell RNA-seq datasets[30]. To this aim, we plotted the percentage of genes in the lists of up- or down-regulated genes expressed in each cell type and overlaid this percentage as a color code to the scRNA-seq UMAP. As for TEA, we found that

genes upregulated upon COH-1 cleavage were mostly expressed in neuronal cell types, while genes downregulated were mostly expressed in the germline (Fig. S10c–f). Altogether, both analyzes strongly suggest that genes upregulated upon COH-1 cleavage are expressed in neurons.

## Cohesin^COH-1 cleavage leads to isoform switch of the nematode Nrf homolog skn-1

We next sought strains in which transcription factors were endogenously tagged and located either close or inside a fountain region, in particular, transcription factors expressed in neuronal cell types. *skn-1/Nrf* is such a gene, as its TSS lies between two very close fountains. The tip of one fountain lies within the gene body of *skn-1* whereas the other fountain lies about 5 kb upstream of the *skn-1* gene, in the *bec-1* enhancer region, and partially merges with the first one and could not be detected by our algorithm as a separate fountain due to their proximity (Figs. S2e, f, and 5a). In RNA-seq data, *skn-1* shows a modest 1.04-fold upregulation upon COH-1 cleavage compared to TEV control. *skn-1* expression has been widely studied, due to its function in lifespan regulation (for review[31]). SKN-1 has three isoforms named A to C, arising as a result of the use of different TSS and alternative splicing (Fig. 5a). At the third larval stage, the *skn-1c* isoform was undetectable by RNA-seq, hence, we focused on *skn-1a* and *skn-1b*. The SKN-1A transcription factor is present in intestinal cells and normally located in the cytoplasm due to its membrane targeting signal at the N-terminus of the protein. Upon stress, SKN-1A relocates to the nucleus for transcriptional gene regulation[32,33]. In contrast, SKN-1B is exclusively nuclear and solely expressed in two head neurons called ASI (named according to the *C. elegans* neuronal nomenclature: Amphid, Single (cell body position is unique), Type I (ciliated endings); Fig. 5b, TEV control[34]).

At the genomic level, *skn-1a* and *skn-1b* share several exons on the 3′ part of the transcripts, but transcription of the two isoforms is controlled by different promoters. The *skn-1b* isoform is transcribed from its own promoter while the *skn-1a* isoform is part of an operon whose promoter is located upstream of the *bec-1* gene (Fig. 5a). Several active enhancers around the *skn-1* gene have been independently identified in entire animals[6,7] (Fig. 5a, d, detailed in S11). One of these enhancers is located between the first and the second exon of *skn-1a*, although it remains unclear in which cells this enhancer is active (Fig. 5a, red squares labeled D[7] and J[6]). Using a C-terminally GFP-tagged construct which labels all SKN-1 isoforms, we observed that in control animals at the L3 stage, most fluorescence was indeed visible in the nuclei of the two ASI neurons, since the low, diffuse cytoplasmic expression in the intestine is not visible above the autofluorescence of the gut granules, as expected from previous studies ($n = 44$, Fig. 5b, TEV control; ref. 35). This exclusively nuclear fluorescence in ASI identifies SKN-1B as the expressed isoform. Upon COH-1 cleavage, a clear increase of the GFP signal was scored, in line with *skn-1a* upregulation observed by RNA-seq (Fig. 5c, fluorescence quantified in 5e, wildtype). Additionally, we detected in all imaged animals ($n = 70$) some degree of accumulation of the GFP-tagged protein in the neuronal cytoplasm, filling the entire cells in most animals, including the sensory dendrites located in front of the cell body as well as the laterally extending axons (Fig. 5b, cohesin^COH-1 cleavage, fluorescence pattern quantified in S12b). This cytoplasmic localization required COH-1 cleavage as heat-shocked animals with TEV cleavage sites in COH-1 but no TEV expression transgene only showed nuclear fluorescence ($n = 21$, data not shown). The presence of cytoplasmic SKN-1 in ASI neurons has never been observed previously (pers. comm. K. Blackwell, HMS). To further assess the nature of the *skn-1* isoform, we took advantage of the stress-mediated nuclear relocation of SKN-1A. We heat-shocked the animals a first time at the L1 stage to induce COH-1 cleavage, and a second time at the L3 stage, 30 min before imaging. Under those conditions, all imaged animals ($n = 25$) showed no cytoplasmic ASI fluorescence anymore, as expected due to the relocation

of SKN-1A to the nucleus upon stress activation (Fig. 5b, cohesin^COH-1 cleavage + heat-shock). Differential expression analysis of *skn-1* isoforms at the transcript level in entire animals showed that *skn-1a* was indeed upregulated upon COH-1 cleavage (Fig. 5c, $p = 0.02$), though the downregulation of the more lowly expressed *skn-1b* isoform was not significant. Additionally, after the cleavage of COH-1, we observe an increase in the expression of the *bec-1* gene to the same levels as *skn-1a*, as expected since both genes are part of the same operon (data not shown). We conclude that cleavage of cohesin^COH-1 has dual effects: it upregulates *skn-1* expression and induces significant changes in the isoforms expressed. Specifically, there is a transition in promoter usage from *skn-1b* to the operon promoter of *bec-1* and *skn-1a*, resulting in the production of the non-neuronal protein SKN-1A. These findings suggest that the cohesin^COH-1-mediated formation of fountains direct promoter use, most likely by restricting active enhancer activity.

## skn-1b proximal sequences regulate skn-1a upon COH-1 cleavage

This model predicts that deletion of one or more active enhancer(s) should reduce transcriptional activity and/or hinder the switch from *skn-1b* to *skn-1a* upon COH-1 cleavage. We tested this by deleting putative regulatory active regions located close to or inside the *skn-1* gene, guided by gene activity or organism-wide mapping data[6–8,23,36] (Fig. 5d, detailed in S11). One limitation here is that enhancers and fountains were identified in entire animals, hence their specific activity in ASI neurons could not be assessed using Hi-C or ATAC-seq data.

We first deleted two distal regions containing putative active enhancers: the entire *nhr-46* gene, located 100 kb upstream of the promoter of *bec-1/skn-1a* operon (Figs. 5d, and S11a, b; *ubs71*), and a large enhancer-rich region situated 5 kb upstream of the *bec-1* gene (Figs. 5d, and S11a, b, b; *ubs62*). In the *ubs71* deletion strain, *skn-1b* expression was modestly elevated in control animals, but GFP intensity remained comparable to the wild-type locus while the COH-1 cleavage-induced isoform switch was slightly reinforced ($n > 20$ animals for each deletion and condition, here and below, quantification in Figs. 5e and S12b). This indicates that this active enhancer is not required for *skn-1a* activation. The ASI-specific SKN-1::GFP mean intensity in the *ubs62* putative enhancer deletion strain was slightly lower in control conditions (~92% of the wildtype locus strain), but upon COH-1 cleavage, expression was restored to wild type levels (Fig. 5e, *ubs62*). Additionally, strong cytoplasmic GFP localization was fully penetrant in this strain upon COH-1 cleavage (Fig. S12b). These results suggest that neither region is essential for *skn-1b* expression or *skn-1* activation. However, the enhanced signal observed in both deletion strains (Fig. S12b) raises the possibility that these enhancer-containing regions may contribute to the insulation of the *bec-1/skn-1a* promoter from neighboring active enhancers.

Subsequently, we deleted the first intron of *skn-1a* located between the first exon of *skn-1a* and the *skn-1b* TSS, reducing its size from 4347 to 551 bp (Figs. 5d and S11a, b; *ubs73*). The switch from *skn-1b* to *skn-1a* upon COH-1 cleavage remained unchanged (Fig. S12b). However, expression of *skn-1b* as well as increased expression of *skn-1* upon COH-1 cleavage was more variable than in control strains (Fig. 5e). The first intron of *skn-1a* thereby appears to stabilize *skn-1a* and *skn-1b* expression, although it is not essential for their transcription. We next deleted a 959 bp region in the third intron of *skn-1a*, 955 bp 5′ of *skn-1b* TSS and 10 kb downstream of *bec-1/skn-1a* operon promoter, overlapping the *clec-178* gene (Figs. 5d, e, and S11a, b; *ubs72*). *clec-178* is exclusively expressed in coelomocytes, phagocytic cells located in the body cavity, yet DNase-I accessible region conformation capture (ARC-C)[8] shows that the *clec-178* region harbors significant contacts with the *skn-1b* promoter and the first intron of *skn-1a*, suggesting the presence of an active enhancer in this region (Fig. S11a, b, ARC-C). Upon deletion of the *clec-178* region, *skn-1b* expression was marginally reduced in control animals (94% of wildtype locus, Fig. 5e, *ubs72*). However, and in contrast to the wildtype locus or all other deletions, COH-1 cleavage led to a

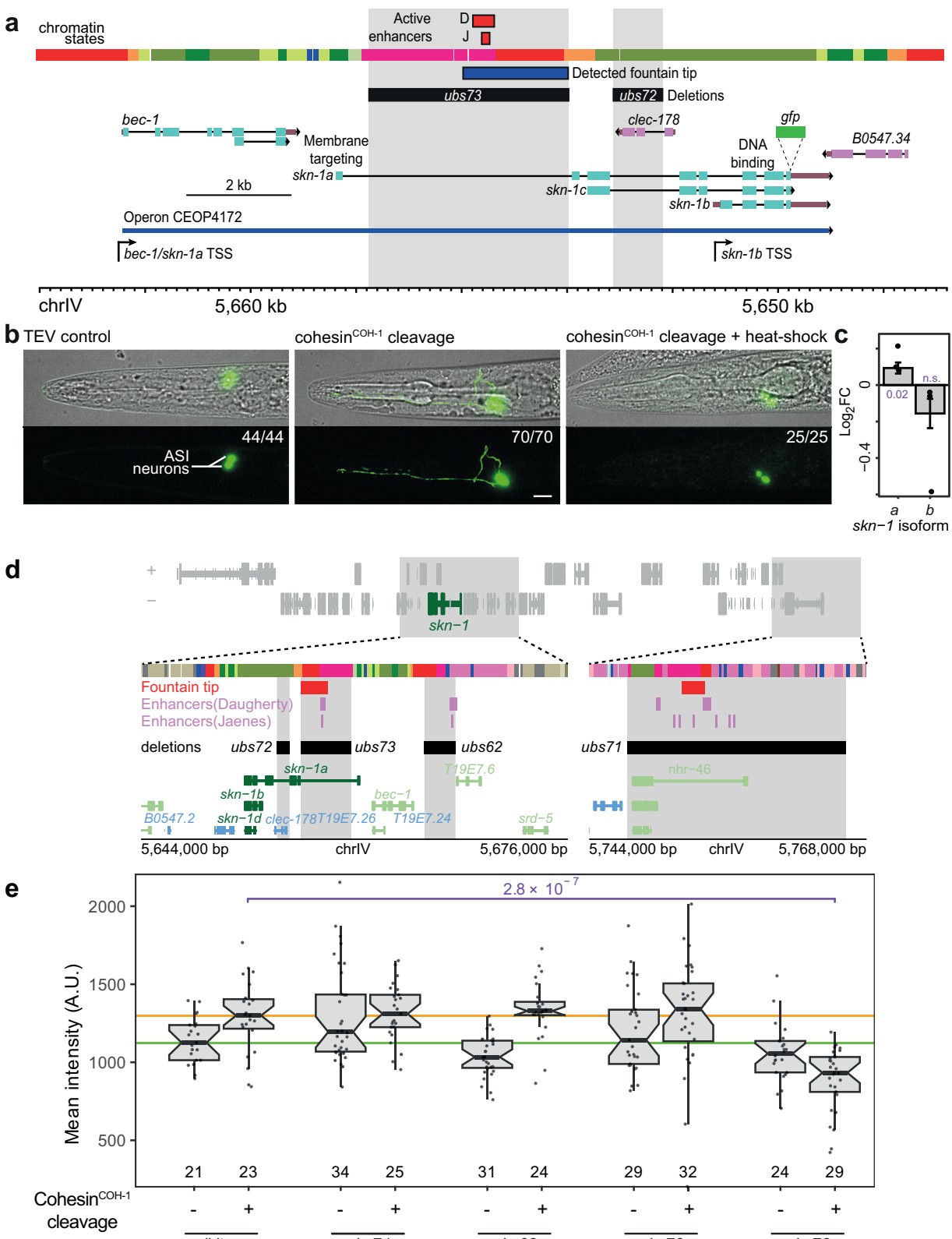

significant downregulation of *skn-1* and reduced switching to *skn-1a* isoform (Figs. 5e, and S12b). Sequences in the deleted region, therefore, contain sequences marginally necessary for *skn-1b* expression in ASI in control animals, but these sequences are required for *skn-1* upregulation and the switch to the *bec-1/skn-1a* operon transcription upon COH-1 cleavage. Our interpretation is that in ASI neurons, sequences in the *clec-178* gene behave as loading sites for cohesin, creating a fountain

which insulates the inactive *bec-1/skn-1a* operon promoter from the transcriptionally active *skn-1b*. Upon COH-1 cleavage and fountain ablation, these regions, as well as other active enhancer regions nearby contact the operon promoter, leading to its activation. Deletion of *clec-178* sequences removes the closest highly active enhancer to *skn-1a* in ASI, leading to decreased use of the *bec-1/skn-1a* TSS upon COH-1 cleavage.

**Fig. 5 | *skn-1/Nrf* switches isoform upon cohesin[COH-1] cleavage. a** Structure of the *skn-1* genomic region, highlighting the three canonical *skn-1* isoforms *a* to *c*, the location of the GFP transgene insertion used in b, the characterized active enhancer (organism-wide), the detected fountain location (organism-wide) and the location of two of the deletions used in (**e**). Active enhancers from Daugherty et al.[7]. (D) or Jaenes et al.[6]. (J). **b** Expression of *skn-1* in the head of control animals, upon COH-1 cleavage or upon COH-1 cleavage and heat stress 30′ prior to imaging. Scale bar: 10 μm. Numbers on the upper left corner of the lower panel indicate the number of animals in which the depicted phenotype has been observed (first number) and the number of animals imaged (second number). **c** Transcript level differential expression of *skn-1* isoforms upon COH-1 cleavage. Adjusted *p* values from a two-sided Wald test are shown next to the bars (n.s.: not significant). Log2FC of transcript levels from the three RNAseq biological replicate batches are shown with dots. **d** Structure of the larger *skn-1* locus, depicting chromatin states (colors as in

Fig. 2g), detected fountain tips, active enhancers, as well as the putative active enhancer deletions. **e** Boxplots of mean fluorescence intensity in ASI nuclei/cell-bodies in control conditions (no cohesin[COH-1] cleavage, −) and upon cohesin[COH-1] cleavage (+) for the different deletion strains depicted in (**d**). The horizontal green and orange lines show the median of the average fluorescence intensity of the ASI nuclei for the wildtype locus without or with cohesin[COH-1] cleavage, respectively. The number of nuclei/cell-bodies in each group is shown at the bottom of the boxplots. FDR adjusted *p*-values from a two-sided Wilcoxon rank sum test comparing wild-type and enhancer deletions strains in control (no TEV expression) and cohesin[COH-1] cleavage conditions are shown in purple. Only significant *p* values are shown. The midline, boundaries and whiskers of the boxplots in **c**, **e** show the median, 25th and 75th percentiles, and the minimum/maximum value no further than ±1.5× the interquartile range, respectively. Notches (in **e**) indicate ±1.58 IQR/sqrt(*n*).

## COH-1 cleavage minimally delays animal growth and development

To further assess the function of cohesin[COH-1], we measured the impact of COH-1 cleavage on animal growth during larval development with a time resolution of 10 min[37] (Fig. S13a). COH-1 cleavage led to a very small difference in animal volume compared to TEV control (Fig. S13b, c). However, this difference was minor compared to previously characterized mutants (*eat-2*, *dbl1*, *raga-1*; ref. 37). We conclude that COH-1 cleavage and fountains disappearance have a minimal effect on animal growth.

## COH-1 cleavage has a broad impact on animal behavior

Since neuronal gene expression is disturbed by COH-1 cleavage (see above), we wanted to analyze its impact on nervous system function using computer-assisted high-content behavioral quantification of worm crawling behavior[38]. We compared 47 core postural and locomotion parameters in wild type (N2), TEV control and upon cohesin[COH-1] cleavage. Animals either dwelled on food or were in search of food after 6 h of deprivation (Fig. 6). We found that COH-1 cleavage affected many behavioral parameters in both conditions, whereas the behavior of TEV control animals was indistinguishable from that of wild type (Fig. S14). Some behavioral differences were quite obvious (Fig. 6a and Supplementary Movie 1, 2). These broad behavioral differences caused animals with cleaved COH-1 to cluster separately in a principal component analysis (PCA) (PC1/PC2 space, Fig. 6b). Among the most salient features, COH-1 cleavage caused a striking increase in the body curvature of both fed and starved animals (Fig. 6a, e) and impaired the ability of worms to implement food search behavior upon food deprivation (Fig. 6c–g). In wild-type animals, the entry into the food search locomotion state involves multiple coordinated behavioral changes[39,40]. These include a marked elevation of animal speed and of the frequency of reorientation events called omega turns (Fig. 6c, d). As a consequence, worms produce relatively twisted trajectories (Fig. 6f) but disperse fast, dramatically increasing worm displacement as compared to fed animals (Fig. 6g). The upregulation of both speed and omega turn frequency upon food-deprivation was strongly reduced upon COH-1 cleavage (Fig. 6c, d), which impaired dispersal (Fig. 6f, g). In contrast, TEV control animals behaved essentially like wild-type ones (Fig. 6a–g). Collectively, these results demonstrate that the post-developmental cleavage of COH-1 produces a broad impact on animal locomotion and are in line with a model in which COH-1-dependent fountain formation is essential for normal neuronal gene expression and the ensuing proper function of the nervous system.

## Discussion

### At least three SMC complexes create looping domains on nematode chromosomes

In mammals, cohesin is the main loop extruder acting on interphasic chromosomes, and its removal leads to the disappearance of most 3D structures including loops and TADs[41–43]. This study, combined with

our previous data, demonstrates that at least three different SMC complexes are at play on interphase nematode chromosomes[9]. On the one hand, large-scale chromosome looping is achieved by condensin I creating loops larger than 100 kilobases, while a variant of condensin I specifically targeted to the X chromosome creates specific loop domains resembling TADs on this chromosome[44]. On the other hand, cohesin[COH-1] achieves short-patch loop extrusion in the tens of kilobases range centered on active enhancers, giving rise to fountains. This is markedly different from Vertebrates, in which different SMC complexes extrude loops during different cell cycle stages, with cohesins and condensins active during interphase and mitosis, respectively[45]. In nematodes, the presence of different interphasic structures maintained by cohesin[COH-1] or condensin I variants is unique, suggesting the simultaneous activity of these complexes, which has not been previously observed in Metazoan species.

Peculiar to nematodes of the *Caenorhabditis* genus, two divergent copies of the cohesin kleisins are present in somatic cells (as well as three additional ones in the germline). Early reports suggest that the two somatic kleisins play different roles, as cohesin[COH-1] is present in somatic cells independently of the cell cycle phase while cohesin[SCC-1] is exclusively expressed in dividing cells[20,46]. Together with previous work in which we tested the mitotic function of the different SMC complexes, our data demonstrate a functional specialization of the two somatic cohesin complexes: cohesin[SCC-1] holds sister chromatids together during mitosis, in line with the observed *scc-1* mutant phenotypes due to chromosome segregation defects in dividing cells[20]. In contrast, cohesin[COH-1] only has a minor role in sister chromatid cohesion, although COH-1 is the most abundant somatic kleisin[9]. The function of cohesin[COH-1] newly described in this report, is therefore small-scale chromatin loop extrusion during interphase, leading to the formation of enhancer fountains. This separation of function between the two cohesin complexes allows the untangling of the mitotic and interphasic functions of cohesin.

### Nematode enhancers correlate with fountain 3D structures

Enhancers play a critical role in transcriptional gene regulation by conferring cell type and developmental stage-specific gene expression. However, their activity must be tightly regulated to prevent unintended activation of non-target promoters[1]. In vertebrates, this regulation is largely mediated by the formation of TADs, established by the combined action of cohesin-mediated loop extrusion and CTCF-bound boundary elements. These 3D chromatin domains restrict enhancer-promoter contacts across domain boundaries[3], thereby minimizing ectopic gene activation. In nematodes, however, high-resolution chromatin conformation capture studies failed to detect TAD structures on autosomes[24,44,47–49], raising the question of how enhancer-promoter specificity is maintained in the absence of TADs. Here we present evidence for the presence of loose, tens-of-kilobase-sized 3D structures centered on active enhancers, which resemble previously described flares, hinges, plumes, or jets in other

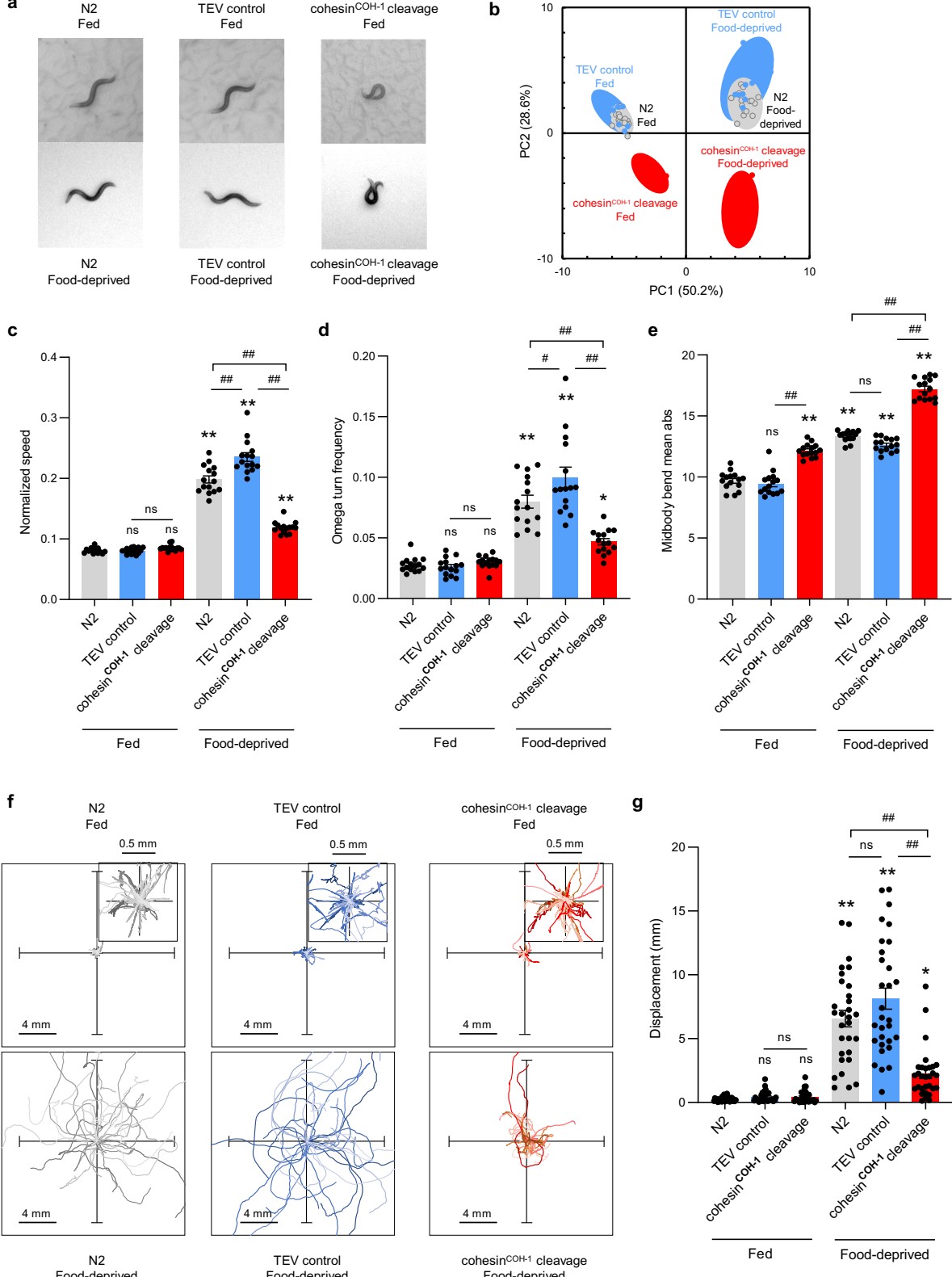

organisms[10–13]. During the preparation of this manuscript, an independent study identified the same cohesin-dependent structures at active enhancers in *C. elegans*, using an orthogonal degron-based acute degradation system targeting the cohesin subunit SMC-3 and the cohesin unloader WAPL-1[50]. Additional experiments will be required to clarify whether fountains at active enhancers arise from altered cohesin dynamics - such as covalent modification of cohesin subunits

or regulation by auxiliary factors - or whether they can be explained solely by localized loading.

These structures appear to be evolutionarily conserved. Their focal points are marked by H3K27ac - a histone modification associated with enhancer activity - not only in nematodes but also in zebrafish sperm and during ZGA[15], as well as in Wings apart-like (WAPL)/CTCF-depleted mouse embryonic stem cells and primary thymocytes[10]. As in

**Fig. 6 | COH-1 cleavage affects the nervous system function. a** Representative pictures of young adult *C. elegans* either dwelling on food (fed) or foraging off-food 6 h after food deprivation (food-deprived), illustrating postural differences between the indicated genotypes. **b** Multidimensional behavioral states of animals presented as projections over the two main principal components (PC1 and PC2) from a single principal component analysis (PCA) over 47 postural and motion parameters, and over all the conditions. The proportion of variance explained by each PC is indicated in the axis labels. Upper left and upper right quadrants correspond to the normal dwelling and food search behavioral states adopted by wild type or TEV control worms. The lower left and right quadrants correspond to markedly altered behavioral states caused by COH-1 cleavage. Averages positions of individual replicates as data marks, 95% CI as colored ellipses. Each replicate is a separate population with ≥40 animals. **c–e** Selected behavioral parameters reported as average ± s.e.m. of *n* = 15 independent replicates. Each replicate value (data points) corresponds to the average value of a separate population with ≥40 animals. Additional behavioral alterations are presented in Fig. S14. **f** One-minute worm trajectories (35 for each condition) plotted from a single starting (0,0) coordinate. Enlarged representations for fed worms in dwelling state (insets). **g** Dispersal quantification. Average ± s.e.m. and individual data points for animal displacement (corresponding to how far animals moved from their starting point). *n* = 30 animals. **c–e**, **g** $^*p < 0.05$ and $^{**}p < 0.01$ versus N2 fed condition, $^#p < 0.05$ and $^{##}p < 0.01$ versus the indicated control by Bonferroni post hoc tests (two-sided tests).

nematodes, ZGA fountains in fishes colocalize with active enhancers[11,15].

In thymocytes[10], the projection angle of jets is modulated by the presence of proximity of CTCF boundaries, resulting in asymmetric loop extrusion. Similarly, nematode fountains can be asymmetrical (Fig. 2h). However, as *C. elegans* lacks a CTCF homolog, this suggests that alternative genomic or chromatin-based barriers may constrain directionality. Notably, genes located on the more-extruded side of asymmetric fountains exhibit lower expression compared to those on the opposite side (Fig. S8), indicating a possible impact of transcription. This difference between enhancer sides is more likely the cause of fountain asymmetry than a consequence of asymmetric fountain formation as removal of fountains by COH-1 cleavage did not change transcription level asymmetry (Fig. S8). Conversely, transcription inhibition did not significantly modify asymmetric fountains. Instead, chromatin states associated with enhancers are more enriched on the more-extruded side, suggesting that these states might be more permissive to loop extrusion. A similar interplay between transcription and loop extrusion has been observed in Bacteria and WAPL/CTCF-depleted mammalian cells, where loop extrusion is impeded by polymerases, while SMC complexes might be pushed by transcribing complexes[19,51–53].

Our analysis of published RNA polymerase II and topoisomerases ChIP-seq data showed that active enhancers are enriched for these factors (Fig. 3b, c; ref. 24). RNA polymerase presence at fountain tips aligns with known bidirectional transcription at active enhancers, while topoisomerase enrichment suggests the accumulation of torsional stress and supercoiled DNA[6]. Indeed, bTMP mapping revealed specific enrichment of negatively supercoiled DNA at active enhancers as well as fountain tips. This implies that, in addition to forming chromatin loops, cohesin may generate supercoils of plectonemes during fountain formation, as observed in vitro[25]. Fountains may also act to locally restrict the dissipation of torsional stress generated by enhancer transcription or by the loop extrusion activity of cohesin loaded onto enhancers[22]. Degron-mediated degradation of topoisomerases I or II only slightly modified fountain strength (Fig. 3e). Topoisomerase I depletion resulted in increased contact frequencies at the fountain tip (the active enhancer locus) and generally stronger fountains, suggesting that topoisomerase I reduces local compaction. Conversely, topoisomerase II depletion led to reduced contact frequencies along the fountain trunk, consistent with a more relaxed or extended fountain structure. Transcription inhibition using α-amanitin generally slightly weakened fountains, but did not ablate them. Fountain maintenance, therefore, appears to be largely independent of topoisomerase activity or transcription, suggesting that once set up, fountains no longer need transcription to remain stable.

### Fountains as potential repressors of active enhancer-proximal gene activity
Our results provide correlative evidence for a link between fountain structures and enhancer-mediated transcriptional regulation. Cleavage of cohesin[COH-1], leading to the disappearance of fountains, is accompanied by transcriptional upregulation of genes located near fountain tips and active enhancers as well as by changes in transcript isoform usage (Figs. 4 and 5). This transcriptional change was not observed in a previous study employing acute SMC-3 degradation[14], likely due to the shorter timeframe between cohesin depletion and sample collection in that study (1-h degradation versus 19-h induction of cohesin[COH-1] cleavage in our system). Given the inherent stochastic nature of enhancer-promoter contacts and transcriptional activation (for review, see ref. 54), it is plausible that changes resulting from fountain loss require more time to manifest at the transcriptional level. Our findings would support the notion that fountains act as repressors of active enhancer function. Upregulation of genes at fountain tips may result from multiple, spatially distinct mechanisms. Locally, the loss of cohesin may alleviate physical interference with the transcriptional machinery, reducing collisions between elongating RNA polymerases and cohesin complexes[19,55]. This model is supported by previous work showing transcriptional repression caused by repeated cohesin passage in mammalian cells following artificial targeted cohesin loading[4,56], and may be especially relevant at fountain tips, where RNA polymerase II density is high (Fig. 3). Such a mechanism could represent a form of enhancer self-regulation: by recruiting cohesin to fountain tips, active enhancers may impose a feedback constraint on their own activity through increased polymerase collisions.

Alternatively, upregulation of the nearest gene could result from enhancer mistargeting. Given the compact nature of the nematode genome, enhancers are often assumed to primarily regulate the closest gene. However, the observed loss of active enhancer-promoter contacts for promoters ranked 2–5 by distance (Fig. S9) suggests that these may represent the actual target promoters. In this scenario, cohesin facilitates longer-range enhancer-promoter contacts, as observed in mammalian cells[4], and when cohesin is lost, polymer dynamics may redirect enhancers to the physically closest promoters. This is particularly intriguing given the enrichment of neuronal genes at fountain tips, which are known for their isoform expansion[57], where choosing between different promoters of the same gene in different cell-types may require cohesin-fountains, as appears to be the case for *skn-1* (Fig. 5).

Beyond local interference, more global changes in chromatin architecture may also contribute to transcriptional upregulation upon cohesin removal. In mammalian systems, cohesin depletion has been shown to increase spatial mixing between accessible chromatin domains[58]. Similarly, in *C. elegans*, fountains collapse following cohesin[COH-1] cleavage, with fountain trunks forming fewer contacts with the enhancers at fountain tips (Fig. S9) and increased contacts with surrounding genomic regions (Fig. 4g). In this context, active enhancer-containing fountain tips may become more mobile, gaining the ability to form new regulatory interactions. This could enhance transcription by promoter trans-acting enhancer-promoter contacts and increasing regulatory crosstalk.

### Cohesin-mediated gene regulation is evolutionarily conserved
Our data reveal that genes overlapping fountain tips exhibit greater structural complexity, characterized by multiple TSS, TES and/or

alternative exons - hallmarks of greater regulatory complexity typically associated with enhancers. The integrity of cohesin[COH-1] thus emerges as a critical factor in regulating such complex genes, as illustrated by the isoform switch observed for *skn-1* (Fig. 5). Remarkably, our investigation revealed a noteworthy upregulation of genes associated with neuronal function following cohesin[COH-1] cleavage (Fig. S10). This conserved role of cohesin in interphase neuronal gene regulation transcends species boundaries. In *Drosophila*, cleavage of the cohesin kleisin in post-mitotic neurons results in impaired axonal and dendritic pruning, as well as abnormal larval locomotion[59]. Similarly, in humans, mutations in cohesin or cohesin-associated genes have been implicated in cohesinopathies, a group of neurodevelopmental disorders[60]. The most well-known of these conditions is Cornelia de Lange syndrome (CdLS), associated with mutations in cohesin subunit NIPBL or other SMC complex subunits[61,62].

In nematodes, cleavage of COH-1 and subsequent loss of cohesin[COH-1] produced a spectrum of behavioral phenotypes (Figs. 6, and S14). Notably, in the absence of food, the animals exhibited altered foraging behavior, displaying sluggish movement instead of active foraging (Fig. 6). This finding is reminiscent of genetic screens in nematodes that identified mutations in the *mau-2* gene (MAternally affected Uncoordination (*mau*)). *mau-2* mutants display defects in axon guidance leading to behavioral phenotypes[63], and the gene gave its name to the mice and human Scc4/MAU2 gene necessary for cohesin function. Our study strongly suggests that the observed *mau-2* phenotypes in nematodes are a consequence of cohesin[COH-1] malfunction in neurons. Similarly, the global dysregulation of genes observed in human CdLS patients further supports the connection to cohesinopathies[64]. Comparison of genes modulated in post-mortem brain neurons of CdLS patients with transcripts modulated upon cohesin[COH-1] cleavage in nematodes using gene set enrichment analysis, revealed a significant overlap for upregulated genes between the two species (Fig. S15; ref. 64). The nematode orthologs of genes upregulated in CdLS were significantly enriched among genes upregulated upon COH-1 cleavage (Fig. S15c), whereas the orthologs of downregulated transcripts in CdLS patients did not display the same pattern (Fig. S15b). Collectively, these similarities strongly indicate that cohesin plays a role in regulating gene expression levels in neurons, most likely through the formation of 3D insulating structures. Further comprehensive studies are required to systematically characterize dysregulated neuronal genes and assess neuronal function upon cohesin[COH-1] cleavage in *C. elegans*.

In summary, our study provides compelling evidence establishing cohesin[COH-1] as the key interphase cohesin in *C. elegans*, with a crucial involvement in neuronal function through the formation of insulating 3D structures. Leveraging the extensive repertoire of genetic tools available in nematodes, further exploration of cohesin[COH-1] holds great promise as a genetically tractable model for investigating human cohesinopathies.

## Methods
### General worm growth and collection
A large unsynchronized worm population of the genotype of interest was grown on four 140 mm peptone-rich plates for about 3 days to get gravid adults. The worms were bleached, and the eggs hatched overnight in the M9 buffer without food. The synchronized L1s were plated on four 140 mm NGM plates (80,000–100,000 worms per plate) and left to grow at 22 °C for 24 h. For specific conditions requiring TEV expression, synchronized L1 animals were left on NGM food for 3 h, before heat-shocking them at 34 °C for 30 min. After 19 h, the L3s were washed from the plates using M9 and did a couple of washes to remove bacteria before proceeding with downstream experiments. For the genomic DNA experiments, the L3s were resuspended in 20 ml of ice-cold M9 and 20 ml of ice-cold 60% sucrose, shaken vigorously, and gently layered

4 ml of ice-cold M9 on top. 30 ml of the supernatant (on top of the sucrose), now containing floating worms, is aspirated and distributed into 50 ml Falcon tubes that were subsequently filled to 50 ml with ice-cold M9. Spun at 200 g for 1 min at room temperature, discarded the liquid, and washed the worms with ice-cold M9 and once with room-temperature M9. The worms were left at room temperature for 25 min for the remaining bacteria in the worms' gut to get digested, washed once with M9, and removed liquid before proceeding to DNA isolation. For Hi-C, RNA-seq and behavioral data presented here (Figs. 1, 2, 3f, 4, 5, and 6), we used heat-shocked animals expressing the TEV protease in the absence of TEV cleavage site as controls (labeled TEV control).

### RNA-seq
RNA sequencing libraries were made by Novogene before sequencing using Illumina NovaSeq (paired-end (PE), 150 bp read length). Reads were aligned to the WS285 transcriptome with Salmon (v.1.9.0) in quant mode with the flags --validateMappings --seqBias --gcBias. Counts per gene were compared with DESeq2 (v1.36.0). GO term enrichment was performed using command line versions of the Wormbase TEA tool[29] and WormCat[65]. The COH-1[cs] RNAseq data were taken from ref. 9 but reanalyzed on its own comparing just the COH-1[cs] (PMW828) strain to the TEV-only control strain (PMW366), aligned to the WS285 transcriptome and pre-filtered before DESeq2 analysis to only include samples with at least 10 reads in half the samples. The different pipeline yielded slightly different results tables, but correlation of the shrunken log2 fold change was 0.97. Of a total 9180 genes left after filtering out genes that oscillate during development[66,67], 895 were significantly upregulated and 993 downregulated (adj$P$<0.05), but as noted previously[9], most of the log2 fold changes were extremely small, with only 98 genes having a LFC > 0.5 and 52 genes with a LFC < −0.5. Transcript level quantification was carried out by mapping reads to transcripts using Salmon as described above, but adding 100 bootstrap samples. The counts were converted to hd5 format with Wasabi (version 1.0.1), and differential expression was carried out with Sleuth (version 0.30.1), filtering out low count transcripts with the default basic_filter, and also removing transcripts from oscillating and non-protein-coding genes by their ids. Scripts used for mapping the RNAseq data can be found at https://github.com/CellFateNucOrg/Bolaji_Luthi_RNAseq/releases/tag/v1.0 [https://doi.org/10.5281/zenodo.17582094] and scripts used to produce some of the plots for this paper can be found at https://github.com/CellFateNucOrg/Luthi_etal/releases/tag/v1.0 [https://doi.org/10.5281/zenodo.17582103].

### Nuclei Isolation
400,000 L3s worms were washed with cold nuclear isolation buffer (NIB) (250 mM sucrose, 10 mM Tris-HCl (pH 7.9), 10 mM MgCl$_2$, 1 mM EGTA, 0.25% NP-0.4, 1 mM DTT, and protease inhibitors). Spun for 1 min in a microcentrifuge, discarded the supernatant, and then snap-freeze the reaction tube in liquid nitrogen. The frozen worm pellet was squeezed into a pre-chilled mortar, placed in a ceramic bowl, put the pestle in it, and hit five times with a hammer. Removed the pestle and ground it into powder form with an electric drill. The powder was scraped from the mortar into a new tube. An equal volume of NIB was added to the powder, and the fully thawed mixture was transferred into a Kontes™ 2 ml glass dounce on ice and centrifuged at 4 °C. Dounced the ground-up worms ten times with the "loose" pestle, then ten times with the "tight" pestle, after which it was transferred to a 1.5 ml Eppendorf tube and centrifuged for 5 min at 100 × $g$ in a microcentrifuge. The supernatant was transferred to a new Eppendorf tube, making sure not to take up any worm debris. Added an equal volume of NIB again and repeated the douncing and spinning four more times till all nuclei were collected. The nuclei are then counted using a hemocytometer and fluorescent microscope.

## bTMP-Seq

bTMP was synthesized as described[26] and supplied by Nick Gilbert's research group. Washed nuclei with wash buffer (WB) (10 mM Tris (pH 7.4), 10 mM NaCl, 3 mM MgCl$_2$, and 0.1 mM EDTA) to remove leftover NIB. Centrifuged for 5 min at 845 × $g$ at 4 °C in a microcentrifuge and discarded the supernatant. Making sure to work with 10 million nuclei per 200 µl volume wash buffer, added 4 µl of 30 mg/ml stock solution of bTMP (i.e., 600 µg/ml final concentration) to the sample. The reaction mixture was incubated in the dark at room temperature for 30 min in an Eppendorf tube on a rotator. Transferred the sample into a 96-well plate to increase the surface area and crosslinked for 10 min (8000 energy). Transferred nuclei to a 1.5 ml Eppendorf tube and spun down at 845 × $g$ at 4 °C in a microcentrifuge for 5 min. The supernatant was discarded, and the volume was made up to 600 µl with WB and incubated with 200 µg/ml proteinase K (12 µl of 20 mg/ml stock) at 55 °C for 16 h (overnight). DNA was purified by phenol:chloroform extraction and isopropanol precipitation as follows: Added an equal volume of phenol:chloroform to the sample and transferred it to a 2 ml PLG tube and mixed thoroughly by inversion for 1 min. Spun in a microcentrifuge at room temperature to separate the phases at maximum speed and transferred the top aqueous phase to a new 1.5 ml Eppendorf tube. The phenol:chloroform extraction was repeated once. Added 0.6 volumes of 2-Propanol, mixed by inverting the tube once, and spun at maximum speed for 30 min at room temperature in a microcentrifuge. All the 2-Propanol was discarded by decanting, washed the pellet twice with freshly prepared 70% ethanol, and spun for 1 min at room temperature at maximum speed between the washes. After the last wash, removed all traces of ethanol by inverting the tube and allowed to dry for ~10 min. DNA was resuspended in 41 µl 1× TE buffer (10 mM Tris (pH7.4), 1 mM EDTA) and quantified by Quanti-Fluor® dsDNA System (Promega). The DNA (1–2 µg DNA / 100 µl volume) was fragmented into ~400 bp fragments using the Bioruptor® sonication system (diagenode) at 30 s ON and 90 s OFF at low power setting for 12 min. For the immunoprecipitation (IP), 20 ng of the DNA sample was collected as input and stored at −20 °C. The rest of the sample was made up to 1 ml with 1× TE buffer. In a 1.5 ml Eppendorf, 1 ml 1× TE buffer and 50 µl avidin conjugated to magnetic beads (Dynabeads™ MyOne™ Streptavidin C1) were added. The mix was rotated at room temperature for 5 min and separated from the beads from the wash on a magnetic stand, after which the clear liquid was discarded; this wash step was repeated twice with 1× TE buffer. Added the 1 ml DNA sample to the washed beads and rotated overnight at 4 °C. Placed the reaction tube on a magnetic stand and removed and discarded the clear liquid. Washed the DNA-bound beads with 1 ml of TSE I (20 mM Tris-HCl pH 8.1, 2 mM EDTA, 150 mM NaCl, 1% Triton X-100, and 0.1% SDS) and mixed by rotating at room temperature for 5 min, separated the beads from the wash on a magnetic stand after which the clear liquid was discarded. Subsequent washes were done with TSE II (20 mM Tris-HCl pH 8.1, 2 mM EDTA, 500 mM NaCl, 1% Triton X-100, and 0.1% SDS), TSE III (20 mM Tris, pH 8.1, 0.25 M LiCl, 1 mM EDTA, 1% NP-40 and 1% deoxycholate (DOC)) and 1× TE buffer as done for TSE I. The IP DNA was eluted by adding a 50 µl elution mix (10 mM EDTA and 95% formamide) and heated at 95 °C for 10 min, vortexing the sample every 2 to 3 min. Placed the reaction tube on a magnetic stand and transferred the clear elute to a new 1.5 Eppendorf tube. Added 150 µl Milli-Q water to make up to 200 µl and purified the DNA according to the SPRIselect User Guide for SPRI-based size selection. 20 µl of Milli-Q water preheated to 70 °C was used for the final DNA elution, and transferred the elute to a new tube and quantified the DNA by Qubit. The Illumina library was then made according to the NEBNext® Ultra™ II DNA library Prep kit for illumina® instruction manual and followed the standard protocol. The libraries were then sequenced at the Next Generation Sequencing platform of the University of Bern on Illumina NovaSeq 6000 (single end (SE) 50 bp read length). The sequence data can be found at the European Nucleotide Archive under accession PRJEB102574 (ERP183965).

## bTMP-seq data analysis

The data generated were preprocessed as follows: Adapter was trimmed using Trim Galore! Version 0.6.6, after which the reads were aligned to the *C. elegans* reference genome version ce11 using Burrows-Wheeler Aligner version 0.7.17. The aligned reads were then sorted based on their genomic coordinates using Samtools version 1.8. Sequencing duplicate reads were removed using Picard version 2.21.8, after which blacklisted genes regions and multi-mapping reads were removed using Samtools. For each control and treatment replicates per experiment, the reads were subsampled to the lowest number of reads (≥5 million reads), after which the read length was elongated to fragment length by extending reads to 200 bp from the start. The coverage was normalized to the number of million reads (RPM) by adding mean coverage. The IP data was then normalized to input data using the ratio operation in deepTools version 3.5.1. Next, the input normalized data were normalized to input normalized genomic DNA data using the subtract operation in deepTools. To make a direct comparison across all experiments, the data were Z-Score normalized using a custom R script (R version 3.6.1). Replicates were averaged using WiggleTools version 1.2.2. The profile plots were then made using deepTools.

## Public datasets analysis (ChIP-seq, ATAC-seq, enhancer types)

All normalized publicly available ChIP-seq and ChIP-chip data sets were lifted to the *C. elegans* reference genome version ce11, after which the replicates were averaged, and profiles were plotted using deepTools. For the segmentation of enhancer types from Jänes et al., 2018 into active or repressed enhancers and H3K27me3-enriched regions, we used the ChromHMM chromatin states from Daugherty et al. 2017.

## Hi-C analysis

Data from refs. [24,49] was processed using HiC-Pro at 2 kb resolution and converted into balanced cool and mcool formats using cooler[68]. Hi-C data pileups and corresponding figures were created using the ARA function of GENOVA in R[69]. To account for the difference in resolution between Hi-C data (2 kb) and enhancer/TSS/HOT sites mappings (1 bp), the genome was tiled into 2 kb bins. Enhancer locations with their annotation (active/repressed/H3K27me3-covered) were taken from refs. [6,7], lifted to ce11 and mapped to 2 kb bins. Consecutive or overlapping bins of the same type were merged before performing pileups. The overlap between non-identical type enhancers was minimal, with only 8, 98 and 34 active/H3K27me3-covered, active/repressed and repressed/H3K27me3-covered bins, respectively, out of 2130, 1586 and 1483 active, H3K27me3-covered and repressed bins, respectively. The same approach was taken for the ref. [6] enhancer list, except that the enhancer type was determined by mapping the enhancer locations to the ChromHMM chromatin states from ref. [23]. To calculate an expected value for the average contact frequency maps, the pileups were calculated with a shift of 20,000 kb (10 bins) compared to the bin of interest containing the enhancer, and removal of the first and last percentile (0.01–0.99) of the observed/expected ratio values. Pileups were displayed using the "visualize" function of GENOVA. For genome-wide Hi-C data, contact maps were loaded onto HiGlass for data exploration and ratio map creation[70].

## Hi-C fragment analysis of enhancer promoter contacts

Hi-C fragment pairs that were no more than 30 kb apart were extracted from the allValidPairs files produced by the HiC-Pro pipeline. Only fragments that overlapped enhancers or promoters (TSS+ -100bp) were retained and counted. GenomicRanges were constructed from HiC fragments that overlapped each set of enhancers and promoters. If

enhancers or promoters overlapped more than one HiC fragment they were expanded to include neighboring HiC fragments. A Bioconductor GenomicInteractions object was created with fragment pairs overlapping the constructed GenomicRanges for enhancers and their 10 closest promoters. Counts were normalized by the Hi-C library sizes, and the ratio of COH-1 cleavage / TEV only control counts was calculated. Gene-level RNAseq log2FoldChange values were assigned to all TSSs of that gene, as transcript level estimates were not always well defined. For plotting, only enhancer-promoter pairs that had >10 counts in both the TEV control and COH-1 cleavage conditions were considered. Custom scripts are available at https://github.com/CellFateNucOrg/Luthi_etal/tree/main/fragmentMapping [https://doi.org/10.5281/zenodo.17582094].

### Inferring fountain positions, lengths and asymmetries

We base our fountain detection algorithm on a standard method used in computer vision: a mask representing an idealized version of the object or motif to detect inside an image is locally convoluted to the image. To each pixel of the image a score of object localization is assigned and maxima or peak detection algorithm is then applied to find the pixels where the motifs are localized. In the context of fountain detection, the image is a Hi-C related matrix and the mask is representing a fountain.

More concretely, to focus on cohesin-dependent fountains (i.e., to exclude other motifs that may generate false positive detection and to detect fountains that may be hindered inside other strong cohesin-independent motifs), we used the matrix $M$ of log10 ratio between the TEV control and cohesin$^{COH-1}$-cleaved Hi-C maps as an input image (Fig. 2F). Based on a simple polymer model of loop extrusion, we defined an idealized fountain mask $F(\alpha_-, \alpha_+)$ where $\alpha_-$ and $\alpha_+$ are related to the left and right extrusion speeds respectively and control the extent and asymmetry of the fountain (Fig. S3). First, we considered a symmetric mask $\alpha_- = \alpha_+ = \alpha_O$ and chose $\alpha_O = 0.35$ to have a similar fountain size than those observed around enhancers (Fig. 1). This would correspond to an average extrusion speed of ~4.6/T kbp/min, with T the average residence time in minutes of extruding cohesins on chromatin. We then compute for each position $i$ along the genome the element-wise product matrix $fm(i)$ between $F(\alpha_O,\alpha_O)$ and $M$ centered around $i$. We define the fountain score $S(i)$ as the mean value of the $fm(i)$ elements (Fig. S1a–c). We used the $find\_peak$ function of python package $scipy$ to detect the peaks in the $S(i)$ profile and compute their prominence $P(i)$. These peaks represent putative fountain positions. To select statistically significant peaks, we generate a null model for each chromosome by computing several random matrices by randomly shuffling the subdiagonals of the original matrix $M$. By repeating the fountain score and peak detection operations on these matrices, we obtain a list of 'random' prominences as an empirical null model. Peaks obtained from $M$ with a prominence $P(i)$ larger corresponding to a $p$-value of the null model lower than 0.05 were then selected as significant. To detect for possible asymmetry in fountain shapes that may alter the inference of the exact fountain position, we finally scanned around the approximated fountain position for the presence of asymmetric fountains by computing fountain scores for various asymmetric binarized masks $F(\alpha_-, 2\alpha_O - \alpha_-)$ with $\alpha_- \in [0.05:0.05:2\alpha_O - 0.05]$ ranging from left- to right-handed shapes (Fig. S3). From this, we estimate an asymmetry score ($\in [-1,1]$) representing the degree of asymmetry of the fountain (<0: left-handed fountain corresponding to faster extrusion upstream of the fountain tip (5′ side of the tip), >0: right-handed fountain corresponding to faster extrusion downstream of the fountain tip (3′ side of the tip), Figs. 2H, and S1). For example, the first and fifth quintiles of asymmetry score (Fig. 2H) have a median score of ~±0.4, respectively, corresponding to a difference in extrusion speed of ~2/T kbp/min between the upstream and downstream regions around the fountain origin. To estimate the length of each fountain, we consider the average value $<M(d)>$ of the log-ratio matrix in a band of width $w$ and size $d$ perpendicular to the main diagonal and centered around the fountain origin. The fountain length $l$ is then defined as $(2d_{opt} + 1) \times 2$ kb with $d_{opt}$ the size $d$ for which the difference between $<M(d)>$ and the average value of M at distances larger than $d$ is maximal (Fig. S1). Note that similar mask-based strategies are also employed by other Hi-C-motif-detection methods such as ChromoSight[71] or fontanka[15]. Results from our method and those obtained with fontanka are very consistent (Fig. S1d–f). All the details about mask definition, computation of fountain and asymmetry scores as well as estimation of the null model are given in a python $jupyter$ notebook given in Supplementary Data 1 and available at https://github.com/physical-biology-of-chromatin.

### Transcription inhibition experiments

Starting with 400,000 L3s in 10 ml M9 in a 50 ml falcon tube, added 200 μg/ml **α**-amanitin (Sigma-Aldrich) and OP50 that had been inactivated by heating at 65 °C for 30 min. As a control, incubated the same amount of worms and inactivated OP-50 in the same volume with an equal volume of Milli-Q water. The reaction mixtures were incubated, rotating at room temperature in the dark for 5 h. Spun the worms at $201 \times g$ for 1 min; the liquid was discarded, then washed the worms once with M9 and spun down at $201 \times g$ for 1 min. Resuspended the worms in 100 μl M9 with 200 μg/ml **α**-amanitin (for control, added Milli-Q water instead).

### Fluorescent imaging of nematodes

Bleach-synchronized L1s were grown on OP-50 for 3 h at 22 °C and then heat-shocked at 34 °C for 30 min to induce TEV expression and COH-1 cleavage. L3 animals were collected 19 h post induction and anesthetized using 200 μl of 0.25 M levamisole. Alternatively, worms were grown at 20 °C, heat shocked 3 h after seeding L1s, and L3s collected 24 h post induction. No difference in phenotype was observed between these two regimes. Worms were transferred on a 2% agarose pad and imaged with a Nikon Ti2 microscope equipped with a Yokogawa spinning disk at 25 °C. Z-stacks spaced by 0.3 μm covering 15 μm were acquired for GFP (488 nm), and the brightfield channel and maximum projections were merged in Fiji[72].

To evaluate the ectopic expression of $skn-1a$ in ASI cells, a custom script in python was used to perform a maximum intensity projection of the Z-stacks and determine the 0.5 and 99.5th quantiles of the GFP signal in each of the SKN-1::GFP images without any enhancer deletion. The average of these values was then used as lower and upper bounds of the signal for all the Z-projected images in the data set. Files were saved with randomly assigned anonymous names and scored blindly by two separate individuals (see scoring scheme in Fig. S11a). The average of the two scores for each worm was used. To quantitatively assess the SKN-1::GFP signal in ASI nuclei and cell-bodies proximal to the nucleus, the fluorescence signal was segmented using otsu thresholding. Segmented objects were size selected (min = 500, max = 6000) to remove fluorescence from gut granules or other objects. One image with more than 4 objects (expect to see 1-2 ASI from 1-2 worms) was removed due to excessive background signal. The quality of segmentation was confirmed by manual inspection. The binary segmentation was used to measure mean fluorescence of the ASI nuclei/cells.

### Microchambers preparation and growth analysis

Eggs from bleach synchronized parental culture were transferred into agarose-based arrayed $600 \times 600 \times 20$ μm microchambers[37], mounted onto a 3.5 cm gas permeable polymer dish (ibidi). Dishes were placed in a custom-made stage holder of a 22 °C temperature-controlled Nikon Ti2 epifluorescence microscope equipped with a Hamamatsu ORCA Flash 4 sCMOS camera. Images were acquired with a $10 \times 0.45$ NA objective every 10 min with 10 ms exposure time to minimize motion blur[73], refocusing using Nikon's NIS software autofocus. Three

hours after the start of time lapse imaging, the dish holder was removed from the microscope and dishes were subjected to a 30 min heat shock at 34 °C to activate TEV expression. The dish holder was then mounted back onto the microscope and time lapse imaging restarted 75 min after dish removal. For the analysis, animals were selected which had hatched at least 150 min before heat-shock induction. Volume trajectories were determined using automated image analysis[73] and manually curated for mistakes in molt annotation.

## Behavioral analysis

Behavioral analysis was conducted at 22 °C in young adult animals, as previously described[40]. Briefly, animals were synchronized and, except for N2 controls, submitted to a heat-shock to induce TEV expression as described above. Fed animals were maintained on 6-cm NGM plates with a lawn of OP50 bacteria. Food-deprived animals were deposited on unseeded NGM plates after 3 washes with distilled water. Several hours prior to experiments, plates were transferred into a custom-designed temperature- and vibration-controlled recording system. High resolution (2448 × 2048 pixels) 3-min movies of worm behavior (~50 animals/movie) were acquired with a DMK33UX250 camera (The Imaging Source), at 10 frames per second using IC Capture software (The Imaging Source). Movies were analyzed with the Tierpsy tracker (v1.4[38]; and we focused on a subset of 47 interpretable parameters, the choice of which was previously discussed[40]. PCA and hierarchical clustering were performed with Clustvis webtool by default SVDimpute algorithm[74]. Each condition and genotype were recorded on at least 3 separate days with multiple replicates per day.

## Comparative gene set enrichment analysis

Human-nematode orthologous genes were downloaded from Ortholist2[75]. Only orthologs that were detected by at least two programs were used in the analysis. A table of genes differentially expressed in CdLS neurons compared to control neurons[64] was kindly provided by Matthias Merkenschlager. Genes were ranked by their Log2 fold change after COH-1 cleavage, and their ranks used to test for enrichment of worm orthologs of significantly up and downregulated genes ($p$adj<0.05) in CdLS patients using the fgsea package in R.

## Reporting summary

Further information on research design is available in the Nature Portfolio Reporting Summary linked to this article.

## Data availability

The Hi-C and RNA-seq data generated in this study have been deposited in the GEO and ENA databases under accession code GSE199723 and PRJEB102574 (ERP183965). Processed Hi-C and bTMP data is available on Zenodo https://doi.org/10.5281/zenodo.17582193.

## Code availability

Scripts used for mapping the RNAseq data can be found at https://github.com/CellFateNucOrg/Bolaji_Luthi_RNAseq/releases/tag/v1.0 [https://doi.org/10.5281/zenodo.17582094] and additional scripts used to produce plots for this paper can be found at https://github.com/CellFateNucOrg/Luthi_etal/releases/tag/v1.0 [https://doi.org/10.5281/zenodo.17582103].

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

## Acknowledgements

We would like to thank Cihan Elcin, Laurence Bulliard et Lisa Schild for technical help, Dr. Pamela Nicholson and her team at the NGS platform of the University of Bern, the Meister laboratory for discussion. Some data were acquired on machines supported by the Microscopy Imaging Center (MIC) of the University of Bern. Some strains were provided by the CGC, which is funded by NIH Office of Research Infrastructure Programs (P40 OD010440). This work was funded by the Swiss National Science Foundation 31003A_176226/310030_212472 (to PM), PCEFP3_181204 (to BT), 10030_197607, BSSGI0_155764 and PP00P3_150681(to DAG), the University of Bern and Fribourg, the Agence Nationale pour la Recherche (ANR-18-CE45-0022-01 to DJ), the Wellcome Trust (to NG) and the Novartis Foundation for Medical/Biological Research (to PM).

## Author contributions

B.N.L., J.I.S., A.H., K.G., M.D., P.M. generated strains and performed molecular biology assays and their analyses, B.N.L., J.I.S., and P.M. performed bioinformatic data analyses, N.G. synthesized bTMP and provided guidance for bTMP-seq, S.T. and D.A.G. performed motility assays, K.S. and B.T. measured animal development, D.A. and D.J. analyzed Hi-C data. B.N.L., J.I.S., D.A.G., B.T., D.J., and P.M. wrote the first draft of the manuscript. All authors provided critical feedback and contributed to the final manuscript.

## Competing interests

The authors declare no competing interests.
