## [Transparent Peer Review file · Nature Communications]

Cohesin forms fountains at active enhancers in *C. elegans*

Corresponding Author: Professor Peter Meister

Version 0:

Reviewer comments:

Reviewer #1

(Remarks to the Author)

I thank the authors for addressing the concerns voiced by my previous reviews

With respect to the question whether fountains have a role in gene regulation the authors make the argument that genes located in fountains are upregulated after COH-1 depletion while genes in regions with similar cohesin occupancy outside fountains are not. However, the authors' model that fountains result from local cohesin loading at the fountain tips followed by coordinated cohesin movement implies that cohesin dynamics at fountain regions and non-fountain regions cannot be the same.

With respect to the question how the loss of COH-1 affects enhancer-promoter contacts, the evidence presented points to reduced enhancer-promoter contacts for genes that were highly connected to enhancers in the presence of cohesin. It remains unclear whether this is a chance observation (i.e. if an observable changes it is likely that the direction of change trend towards average rather than extreme values) or is there a systematic relationship between gene position relative to fountain tips, connectivity and expression upon COH-1 depletion?

With respect to the example of a domain-like structure, apologies, I meant to refer to chrIII:3699001-3899000

The authors now propose that fountains act as 'spatial repressors of enhancer activity'. This model implies that COH1 depletion and loss of fountains affects enhancer states through spatial mechanisms. I can see no data to demonstrate enhancer activation following COH-1 depletion, and this model is therefore unsupported. Please make a clear distinction between enhancer state, enhancer-promoter connectivity and gene expression.

As stated in my initial review of this work I believe that the selective formation of fountains at active enhancers is interesting because there are no other known enhancer-specific 3D features, and because the formation of COH-1-dependent fountains at enhancers suggests the possibility that enhancers may be sites of cohesin loading in *C. elegans*. This interest is irrespective of whether fountains have a specific role in the regulation of enhancer states or gene expression. The authors appear to think that this is not enough and continue to push interpretations that remain largely unsupported by the data.

Reviewer #2

(Remarks to the Author)

Luthi et al. have thoroughly addressed the concerns raised in the previous reviews, significantly strengthening the manuscript. In particular, the new analyses convincingly support a link between contact frequency/fountain strength and gene upregulation upon cohesin depletion/fountain disruption. The reorganized presentation of the *skn-1* deletion analysis also improves clarity and ease of interpretation. These revisions collectively enhance the overall quality, conclusions and impact of the study. The revised manuscript is ready for publication.

Minor editorial comments: Please ensure that figure references are updated and supplemental figures are in sequence. For example:

- Page 16, section "COH-1 cleavage minimally delays animal growth and development", update Figure S12 to Figure S13
- Page 16, section "COH-1 cleavage has a broad impact on animal behavior", line 8 of this paragraph, update Figure S13 to Figure S14
- Page 20, discussion, bottom paragraph line 3, update Figure 5 to Figure 6
- Fig. S5 is referenced after later Supplemental Figures

Version 1:

Reviewer comments:

Reviewer #1

(Remarks to the Author)

I thank the authors for revising their manuscript. My remaining concerns have now been addressed.

Rebuttal letter

We thank the reviewers for their constructive feedback on our revised manuscript, and for recognizing the advances made in our analyses. Below, we address each point in detail, referencing new or previously presented results to clarify our interpretations and highlight the robustness of our conclusions.

Reviewer #1 (Remarks to the Author):

I thank the authors for addressing the concerns voiced by my previous reviews

With respect to the question whether fountains have a role in gene regulation the authors make the argument that genes located in fountains are upregulated after COH-1 depletion while genes in regions with similar cohesin occupancy outside fountains are not. However, the authors' model that fountains result from local cohesin loading at the fountain tips followed by coordinated cohesin movement implies that cohesin dynamics at fountain regions and non-fountain regions cannot be the same.

We appreciate the reviewer's insightful observation that our model of fountain formation - *via* local cohesin loading at fountain tips followed by coordinated movement - could imply distinct cohesin dynamics at fountain versus non-fountain loci. We agree that this distinction reinforces the idea that fountains represent unique features compared with genomic regions of similar cohesin occupancy. To reflect this, we have added a discussion point acknowledging the possibility of distinct cohesin dynamics at these sites. At the same time, we feel that directly addressing whether and how cohesin dynamics differ at active enhancers and fountains lies beyond the scope of the present study and will require substantial further experimentation.

Here is the additional sentence, in the discussion, paragraph "Nematode enhancers correlate with fountain 3D structures": "Additional experiments will be required to clarify whether fountains at active enhancers arise from altered cohesin dynamics - such as covalent modification of cohesin subunits or regulation by auxiliary factors - or whether they can be explained solely by localized loading."

With respect to the question how the loss of COH-1 affects enhancer-promoter contacts, the evidence presented points to reduced enhancer-promoter contacts for genes that were highly connected to enhancers in the presence of cohesin. It remains unclear whether this is a chance observation (i.e. if an observable changes it is likely that the direction of change trend towards average rather than extreme values) or is there a systematic relationship between gene position relative to fountain tips, connectivity and expression upon COH-1 depletion?

We thank the reviewer for raising this important point. To circumvent regression to the mean artifacts, our analysis now ranks promoters by their linear distance from a specific active enhancer. This revised methodology provides a clearer picture of the local effect of cohesin^{COH-1} cleavage. Our updated data show that while promoters closest to active enhancers maintain a contact ratio near 1, promoters at slightly greater distance (ranks 2-5) show a subtle reduction in contact frequency. This additional analysis is now integrated into the revised manuscript as Figure S9. However, a direct correlation between contact frequency changes and modifications of the transcriptional levels remains elusive. We attribute this disconnect to the use of whole animals for both Hi-C and mRNA-seq, which averages signals across a diverse population of cells. Coupled to the high density of genes in the nematode genome, this lack of cellular and genomic resolution prevents us from observing cell-specific and promoter-specific regulatory changes. While a more granular analysis is warranted, it is beyond the scope of this study. Despite this, a consistent trend emerges: genes whose promoters are most proximal to active enhancers are upregulated upon cohesin^{COH-1} cleavage.

With respect to the example of a domain-like structure, apologies, I meant to refer to chrIII:3699001-3899000

We would like to thank the reviewer for clarifying which fountain was meant. Here is the picture from supplementary file 3 with the fountain in question.

chrIII:3699001-3899000

At this locus, as in others, the observed configuration matches the general fountain signature, although local active enhancer multiplicity (upper line ticks, right below genes) would broaden the base of the structure. The visual variation is consistent with a complex regulatory landscape and/or variability arising from cell-type heterogeneity in whole-animal Hi-C.

The authors now propose that fountains act as 'spatial repressors of enhancer activity'. This model implies that COH1 depletion and loss of fountains affects enhancer states through spatial mechanisms. I can see no data to demonstrate enhancer activation following COH-1 depletion, and this model is therefore unsupported. Please make a clear distinctions between enhancer state, enhancer-promoter connectivity and gene expression.

We thank the reviewer for this remark. We think that there is some misunderstanding of what we meant by “spatial repressors of enhancer activity”. This was not meant in the way that enhancers are more or less active, but that their enhancer activity is regulated spatially, with regard to the target genes. We have now clarified this sentence in the abstract which now reads “Functionally, fountain disassembly correlates with transcriptional upregulation of genes proximal to active enhancers, raising the possibility that fountains serve as spatial repressors by regulating enhancer–promoter contacts.”

As stated in my initial review of this work I believe that the selective formation of fountains at active enhancers is interesting because there are no other known enhancer-specific 3D features, and because the formation of COH-1-dependent fountains at enhancers suggests the possibility that enhancers may be sites of cohesin loading in C. elegans. This interest is irrespective of whether fountains have a specific role in the regulation of enhancer states or gene expression. The authors appear to think that this is not enough and continue to push interpretations that remain largely unsupported by the data.

We thank the reviewer for highlighting the interest of our findings, in particular the selective formation of fountains at active enhancers and the potential implications for cohesin loading in *C. elegans*. We agree that these observations are noteworthy regardless of whether fountains have a direct regulatory role in gene expression. In response to the reviewer's concern, we have carefully reviewed the manuscript to ensure that our descriptions of the data remain strictly factual and appropriately cautious.

In the abstract, we have adjusted the wording of our conclusions in the final sentence of the abstract: "Together, our findings uncover fountains as a novel 3D chromatin feature that *could* modulate enhancer targeting in a TAD-less genome, establishing a *potential* mechanistic link between genome architecture, gene regulation and behavior."

In the introduction, we describe the "upregulation of active enhancer- and fountain-proximal genes, in particular genes expressed in neurons" as a direct outcome of the RNA-seq analysis (Figure 4).

In the Results, we state that "Cohesin^{COH-1} cleavage correlates with transcriptional activation of genes close to active enhancers and fountain tips." Importantly, we do not claim causality, but instead emphasize the observed correlation, which is consistently supported by the data in Figure 4. We analyzed this correlation from multiple perspectives - considering genes near different enhancer types, genes proximal to fountain tips, and reciprocally - while explicitly noting caveats such as the limited number of genes in some groups. For example: "the number of genes in these groups are small"; "Together, this demonstrates that fountain ablation by COH-1 cleavage correlates with upregulation of fountain tip-proximal genes."

In the Discussion, we have revised the section previously titled "Fountains repress active enhancer-proximal gene activity" to "Fountains as potential repressors of active enhancer-proximal gene activity" to better highlight the speculative nature of these interpretations. In this section, we now present our hypotheses as tentative models, including both a local mechanism (RNA polymerase/loop extrusion factor

collisions, as described in bacteria and mammalian cells, or enhancer mistargeting) and a global mechanism (enhancer clustering in subnuclear domains). Both are framed as possible explanations rather than firm conclusions.

Finally, in the paragraph comparing transcriptional changes across species, we now describe the findings in terms of cohesin integrity rather than fountains, to avoid overinterpretation.

Reviewer #2 (Remarks to the Author):

*Luthi et al. have thoroughly addressed the concerns raised in the previous reviews, significantly strengthening the manuscript. In particular, the new analyses convincingly support a link between contact frequency/fountain strength and gene upregulation upon cohesin depletion/fountain disruption. The reorganized presentation of the *skn-1* deletion analysis also improves clarity and ease of interpretation. These revisions collectively enhance the overall quality, conclusions and impact of the study. The revised manuscript is ready for publication.*

We are grateful for the reviewer's positive assessment and recognition that our new deletion analyses strengthen the mechanistic case for enhancer regulation at fountains.

Minor editorial comments: Please ensure that figure references are updated and supplemental figures are in sequence. For example:

- Page 16, section "COH-1 cleavage minimally delays animal growth and development", update Figure S12 to Figure S13

The figure number has been updated. We apologize for the mistake.

- Page 16, section "COH-1 cleavage has a broad impact on animal behavior", line 8 of this paragraph, update Figure S13 to Figure S14

The figure number has been updated. We apologize for the mistake.

- Page 20, discussion, bottom paragraph line 3, update Figure 5 to Figure 6

The figure number has been updated. We apologize for the mistake.

- Fig. S5 is referenced after later Supplemental Figures

The figure order has been updated. We apologize for the mistake.